**RESEARCH**                                                                                    **Open Access**

# Time-dependent effect of 1,6-hexanediol on biomolecular condensates and 3D chromatin organization

Xinyi Liu[1,2,3†], Shaoshuai Jiang[1,2,3†], Lin Ma[1,2,3], Jiale Qu[1,2,3], Longying Zhao[1,2,3], Xing Zhu[1,2,3] and Junjun Ding[1,2,3,4,5*] 

* Correspondence: dingjunj@mail.
sysu.edu.cn
†Xinyi Liu and Shaoshuai Jiang
contributed equally to this work.
¹RNA Biomedical Institute, Sun
Yat-Sen Memorial Hospital,
Zhongshan School of Medicine, Sun
Yat-Sen University, Guangzhou
510080, China
²Center for Stem Cell Biology and
Tissue Engineering, Key Laboratory
for Stem Cells and Tissue
Engineering, Ministry of Education,
Sun Yat-Sen University, Guangzhou
510080, China
Full list of author information is
available at the end of the article

## Abstract

**Background:** Biomolecular condensates have been implicated in multiple cellular processes. However, the global role played by condensates in 3D chromatin organization remains unclear. At present, 1,6-hexanediol (1,6-HD) is the only available tool to globally disrupt condensates, yet the conditions of 1,6-HD vary considerably between studies and may even trigger apoptosis.

**Results:** In this study, we first analyzed the effects of different concentrations and treatment durations of 1,6-HD and found that short-term exposure to 1.5% 1,6-HD dissolved biomolecular condensates whereas long-term exposure caused aberrant aggregation without affecting cell viability. Based on this condition, we drew a time-resolved map of 3D chromatin organization and found that short-term treatment with 1.5% 1,6-HD resulted in reduced long-range interactions, strengthened compartmentalization, homogenized A-A interactions, B-to-A compartment switch and TAD reorganization, whereas longer exposure had the opposite effects. Furthermore, the long-range interactions between condensate-component-enriched regions were markedly weakened following 1,6-HD treatment.

**Conclusions:** In conclusion, our study finds a proper 1,6-HD condition and provides a resource for exploring the role of biomolecular condensates in 3D chromatin organization.

**Keywords:** 1,6-Hexanediol, Biomolecular condensates, 3D chromatin organization

## Introduction

Multiple proteins and nucleic acids are concentrated as dynamic compartments in living cells. These biomolecular condensates are involved in multiple cellular processes, such as transcriptional control [1–3], stress response [4, 5], quality control of proteins [6], and DNA replication [7]. These condensates are disrupted by 1,6-hexanediol (1,6-HD) that interferes with the weak hydrophobic protein-protein or protein-RNA interactions [8, 9]. It is routinely used to identify condensates in vitro and in vivo since it is not specific to any cell type and does not require genomic manipulation. Several

condensates, including stress granules [10], FUS [9], constitutive heterochromatin [11], super-enhancers [1], and transcription-dependent condensates containing mediators and RNA polymerase II (RNAPII) [3], can be disrupted by 1,6-HD. However, the conditions for 1,6-HD treatment vary considerably, and may even trigger apoptosis [12], aberrant protein aggregation, loss of membrane integrity, cell shrinkage [9], chromatin "freeze," and chromatin hyper-condensation [13]. The experimental conditions that can disrupt a specific type of condensate may not work for the others, or may lead to misleading results owing to side effects. In other words, the parameters of 1,6-HD treatment have not been standardized to explore the relationship between condensates and genomic structures and the relevant biological processes.

The chromatin structure is closely related to cell-fate specification and other biological and pathological processes [14–16]. Chromatin adopts a complex three-dimensional (3D) conformation within the nucleus, which can be hierarchically subdivided into chromosome territories, A/B compartments, topologically associated domains (TADs), and loops [16, 17]. Recent studies show that biomolecular condensates are involved in the formation of these hierarchical structures and overall chromatin organization [11, 18–21]. The relationship between these condensates and 3D chromatin structure can be studied by three major strategies: (1) formation of liquid-like condensates in vivo or in vitro by proteins involved in 3D chromatin organization [1, 3, 22–28], (2) effect of knocking out or mutating key factors of biomolecular condensates on 3D interaction networks [29–31], and (3) artificial induction of condensates at specific genomic loci to restructure chromatin [32]. However, these strategies are largely indirect, fail to exclude the impact of genomic manipulation, or only focus on certain types of condensates, and thus not suitable for global-level analysis. In contrast, 1,6-HD can globally disrupt condensates and is a useful tool to explore the role of biomolecular condensates in 3D chromatin organization in vivo.

Here, we systematically tested the effects of 1,6-HD treatment in different concentrations and time durations to find a proper condition. Subsequently, we mapped the 3D chromatin organization at serval time points before, during, and after 1,6-HD treatment by Hi-C (Fig. 1A). We found that 1,6-HD affected biomolecular condensates and 3D chromatin structure in a concentration and time-dependent manner. Specifically, 2 min exposure to 1.5% 1,6-HD dissolved biomolecular condensates, reduced long-range interactions, strengthened compartmentalization, homogenized A-A interactions, triggered B-to-A compartment switch, and enhanced TAD dynamics, whereas 30 min treatment had the opposite effects. Furthermore, the long-range interactions between condensate-component-enriched regions were also significantly weakened upon 1,6-HD treatment. Our study provides experimental evidence of using 1,6-HD to explore biomolecular condensate-based chromatin folding.

## Results

### The effect of 1,6-HD on biomolecular condensates is time-dependent

Mouse embryonic stem cells (mESCs) were treated with different concentrations (1.5%, 5%, and 10%) of 1,6-HD for varying durations (1, 2, 5, 10, 30 min). While 5% and 10% 1,6-HD drastically reduced the cell viability, 1.5% 1,6-HD did not induce apoptosis even after 30 min of treatment (Additional file 1: Fig. S1A). Accordingly, 1.5% was selected

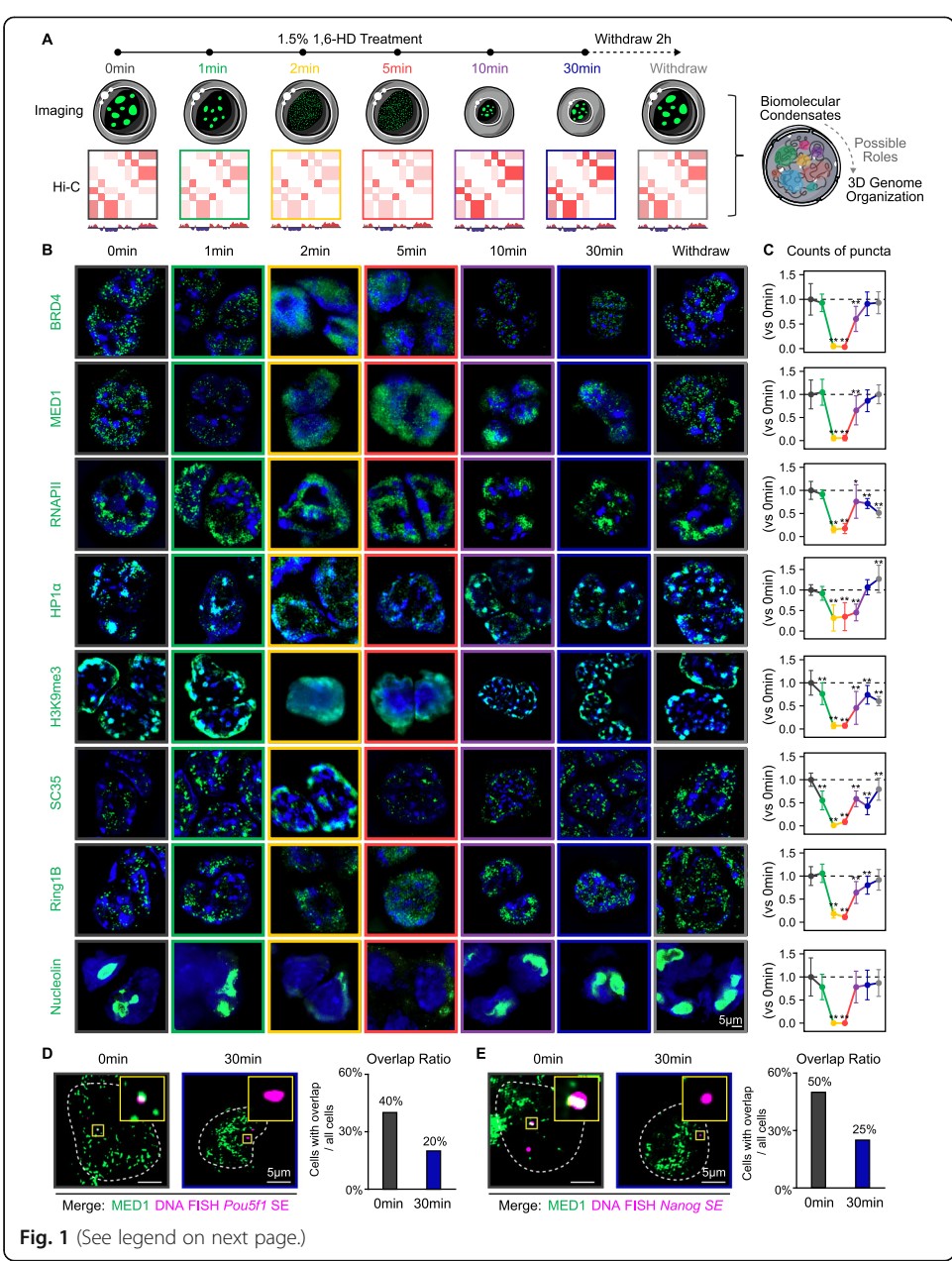

**Fig. 1** (See legend on next page.)

(See figure on previous page.)

**Fig. 1** Short-term 1,6-HD treatment dissolved biomolecular condensates while long-term treatment caused aberrant aggregation. **A** Schematic representation of mouse embryonic stem cells (mESCs) treated with 1.5% 1,6-HD across a time course and after withdrawal. Imaging and Hi-C are performed at each time point to explore the possible roles of biomolecular condensates in 3D genome organization. **B** Structured illumination microscopy (SIM) images of immunofluorescence (IF) for the proteins indicated by green words in mESCs. IF for indicated protein is colored green, and signal from DAPI is colored dark blue. **C** Counts of IF puncta within nucleus at each time points versus 0 min. Error bars represent standard deviation (SD) from 30 cells each. All *p* values were determined using Student's *t* test. **D** Right: merge views of SIM images of MED1 IF with DNA-FISH to the *Pou5f1 SE* locus in mESCs. The white dashed line highlights the nuclear periphery, determined by DAPI staining (not shown). The upper right part shows the area in the yellow box in greater detail. Left: quantification of the proportion of cells with DNA-FISH foci overlapping MED1 puncta. **E** Right: merge views of SIM images of MED1 IF with DNA-FISH to the *Nanog SE* locus in mESCs. The white dashed line highlights the nuclear periphery, determined by DAPI staining (not shown). The upper right part shows the area in the yellow box in greater detail. Left: quantification of the proportion of cells with DNA-FISH foci overlapping MED1 puncta

for the subsequent experiments. Super-resolution imaging of representative biomolecular condensates including super-enhancers (BRD4, MED1) [1], transcription factories (RNAPII) [3], constitutive heterochromatin (HP 1α, H3K9me3) [11, 18], speckles (SC35) [33], polycomb bodies (Ring1B) [22, 34], nucleoli (Nucleolin) [6], and CTCF clusters (CTCF) [28] revealed distinct puncta in untreated cells and after 1 min of treatment with 1.5% 1,6-HD (Fig. 1B, Additional file 1: Fig. S1B). Longer exposure for 2 min and 5 min significantly reduced the number of puncta, while protein reaggregation was induced by 10 min and 30 min treatment. The number of puncta was largely restored after 2 h of 1,6-HD withdrawal (Fig. 1B, C). In contrast, distribution patterns of the proteins that do not form biomolecular condensates, including cohesin (SMC1A) and Lamin B1, were not affected by time-series 1,6-HD treatment (Additional file 1: Fig. S1B). Further, to determine whether such time-dependent effect occurs in live cells, live-cell fluorescence microscopy of endogenous tagged paraspeckles (indicated by NONO) was performed and also revealed the same effect (Additional file 1: Fig. S1C). A previous study showed that the reappearance of protein aggregates is likely due to cell shrinkage and increased macromolecular crowding [9]. Therefore, we next quantified nuclear volume following 1,6-HD treatment and found that the size of the nucleus decreased significantly after 10 and 30 min of treatment (Additional file 1: Fig. S1D). These results indicate a time-dependent effect of 1,6-HD on biomolecular condensates involving condensation, dissolution, reaggregation, and recovery.

Since biomolecular condensates localize at specific genomic loci [28], we determined whether the aggregates induced by long-time 1,6-HD treatment re-accumulated at their targeting regions. As shown in Fig. 1D and E, the proportion of cells with MED1 puncta overlapping with super-enhancers (SEs) of *Pou5f1* gene and *Nanog* gene [23] (as determined by DNA-FISH) decreased significantly after 30 min of 1,6-HD treatment (Fig. 1D, E). Therefore, long-term exposure to 1,6-HD resulted in aberrant reaggregates that fail to localize to their target sites. In addition, previous studies showed that exposure to 1,6-HD decreases chromatin binding of condensate components [1, 35]. We also examined how their binding are affected upon 1.5%, 2 min 1,6-HD treatment, which dissolved condensates. We chose the loci shown as examples in these previous studies [1, 35], where the binding of MED1 decreases dramatically and BRD4 changes moderate. The results showed that 1.5%, 2 min 1,6-HD treatment did not affect the binding of BRD4, but caused a significant reduction of MED1 binding at these loci (Additional file 1: Fig.

S1E), which is consistent with the previous studies [1, 35]. Besides, the binding of CTCF near those loci significantly increased, while the binding of SMC1A remained unaffected (Additional file 1: Fig. S1E). These could be issued with the cluster formation of CTCF [28, 36, 37] and dispersed distribution of SMC1A.

Taken together, 1.5% 1,6-HD induces a time-dependent phase transition of biomolecular condensates with limited toxicity, which is ideal for studying the role of condensates in global 3D chromatin organization.

### 1.5%, 2 min is a proper condition of 1,6-HD treatment across cell types

Notably, treatment with 1.5% 1,6-HD for 2 min dissolved biomolecular condensates without affecting cell viability and nuclear volume, which makes this condition a strong candidate for a proper condition. Since a recent study has reported that 5% or higher concentration 1,6-HD treatment for 5 min had a side effect of suppressing chromatin motion and hyper-condensing chromatin in live cells [13], we next examined the effects of 1.5%, 2 min 1,6-HD on chromatin behavior. We performed super-resolution live-cell imaging on stably expressed histone H2B-GFP in HeLa cells and recorded images at 50 ms/frame (~ 300 frame, 15 s total) (Additional file 1: Fig. S1F). Notably, trajectory analysis revealed that although 10%, 5 min 1,6-HD drastically suppressed chromatin motion as previous reported [13], 1.5%, 2 min 1,6-HD did not affect chromatin motion (Additional file 1: Fig. S1G). Besides, we analyzed the distribution of H2B using the *L* function, which quantitates the size of H2B clusters and the degree of accumulation. As a result, 10%, 5 min 1,6-HD caused chromatin hyper-condensation as reported [13], while 1.5%, 2 min 1,6-HD exhibited a similar H2B clustering to untreated cells (Additional file 1: Fig. S1H).

We also evaluated the effect of 1.5%, 2 min 2,5-hexanediol (2,5-HD), which is structurally similar to 1,6-HD but with less condensate-dissolving activity and was considered as a negative control of 1,6-HD [10, 38]. However, although cell viability and many types of biomolecular condensates remained unaffected upon 2,5-HD treatment, condensates of BRD4 and MED1 were largely dissolved (Additional file 1: Fig. S1I-J). These results suggest that 2,5-HD is not an ideal negative control of 1,6-HD in global-level studies.

We next tested 1.5%, 2 min 1,6-HD in other cell types. Importantly, this condition also dissolved biomolecular condensates without impairing cell viability and decreasing nuclear volume in neural progenitor cells (NPCs) and mouse embryonic fibroblasts (MEFs) (Additional file 1: Fig. S2). These results further support that 1.5%, 2 min is a proper condition of 1,6-HD treatment which is suitable for various cell types.

### Short-term 1,6-HD treatment decreased long-range interactions and increased short-range interactions

The role of biomolecular condensates in overall chromatin folding was determined by mapping 3D genome before and multiple time points (1, 2, 5, 10, and 30 min) after 1.5% 1,6-HD treatment, and 2 h after withdrawal using BAT-Hi-C (Fig. 1A). Two biological replicates were compared at each time point and were highly correlated at all resolutions (Additional file 1: Fig. S3A-C). In addition, the data from untreated cells (0 min) was similar to the ultra-deep in situ Hi-C data of mESCs (Additional file 1: Fig. S3A-S3G). As

shown in Fig. 2A, 1,6-HD treatment induced global reorganization in chromatin interactions (Fig. 2A). Long-range interactions (10 to 100 Mb) decreased significantly after 1–5 min of 1,6-HD treatment, whereas short-range interactions (100 kb to 1 Mb) showed a significant increase (Fig. 2B). In contrast, 10/30 min of exposure increased interactions across 10 Mb and decreased the short-range interactions around 100 kb (Fig. 2B). All interactions were restored after 1,6-HD was withdrawn (Fig. 2B). These effects were also observed in quantification of changed percentage of interactions at certain genomic range in the biological replicates (Fig. 2C), as well as in the representative differential heatmap (Fig. 2D). These results indicate that biomolecular condensates may influence long-range interactions, and different hierarchies of the 3D genome may be differently affected by 1,6-HD treatment.

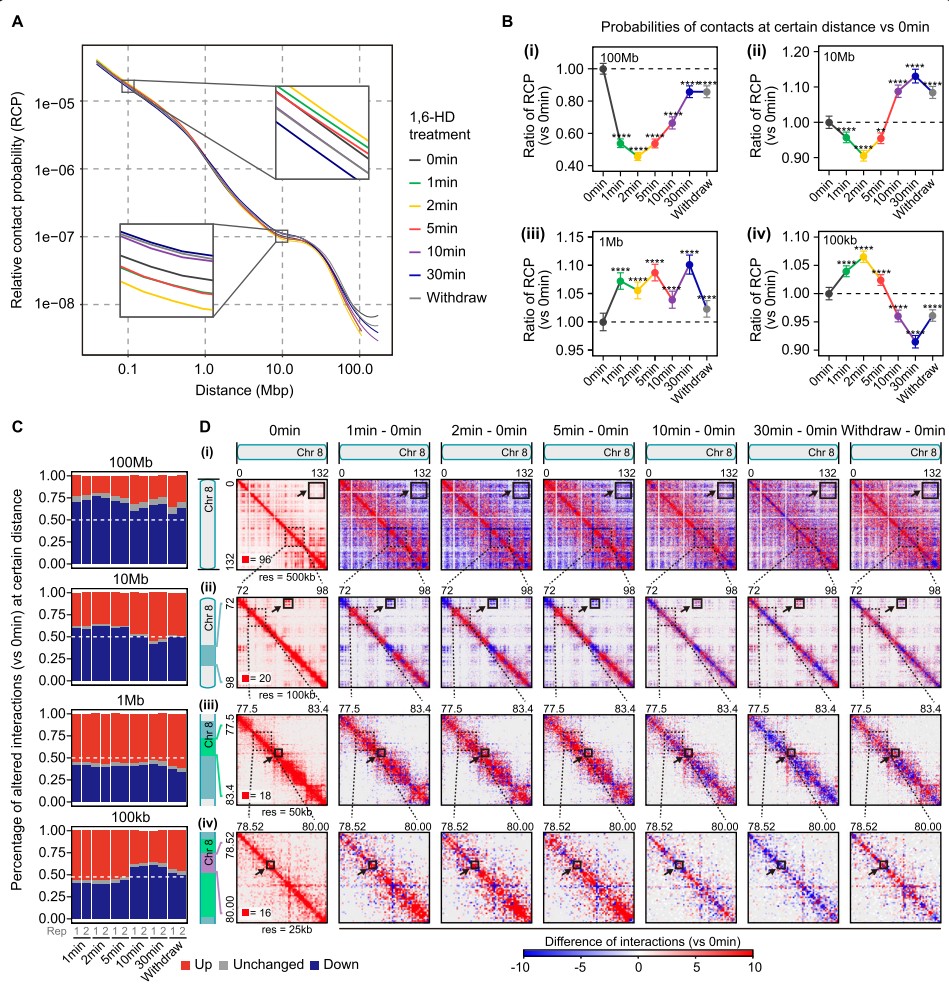

**Fig. 2** Short-term 1,6-HD treatment decreased long-range interactions and increased short-range interactions. **A** Contact probability in logarithmic genomic distance. The two zoom-in boxes show greater details around 0.1 Mb and 10 Mb. **B** Fold change of relative contact probability (RCP) versus 0 min at the distanced annotated on the upper left of each plot. Error bars represent SD from 20 chromosomes (chr1-chr19, chrX). All $p$ values were determined using paired Student's $t$ test. **C** Quantification of changed percentage of interactions at indicated genomic range in the biological replicates compared with 0 min (details in "Materials and methods"). **D** The leftmost column: observed contact matrices for chr8 in untreated cells. Other columns: differential contact matrices for each time point minus 0 min. The bold black line box highlights example regions at distance of 100 Mb, 10 Mb, 1 Mb, and 100 kb respectively

## Short-term 1,6-HD treatment strengthened compartmentalization

Previous studies put forth the hypothesis that biomolecular condensates play a role in partitioning the genome into the open A and closed B spatial compartments [11, 18, 19, 21, 39], and the regions of the same type exhibit enhanced contact frequency. We therefore examined the effects of 1,6-HD treatment on genome compartmentalization using a standard eigenvector-based method that assigned compartments as A or B for each time point, and calculated the average contacts between A-A, B-B, and A-B genomic loci. Surprisingly, short-term (1–5 min) 1,6-HD treatment, especially for 2 min, increased A-A and B-B contacts, decreased A-B contacts and thus strengthened compartmentalization at a global level (Fig. 3A, B, Additional file 1: Fig. S4A-B). The strengthened compartmentalization can also be indicated by the plaid pattern of contact maps observed at 2 min (Fig. 3C). We further performed a per-bin analysis of compartment strength to evaluate the proportion of strengthened or weakened bins. As expected, there were more strengthened bins (with at least 2-fold enhanced compartment strength) than weakened bins (with at least 2-fold reduction in compartment strength) after 1–5 min of 1,6-HD treatment (Fig. 3D). In addition, we compared the compartmental changes between 1.5%, 2 min 1,6-HD treatment and CTCF depletion, since CTCF depletion does not affect compartmentalization [40]. As expected, while contact maps and compartment signal (PC1) were maintained after CTCF depletion, 1.5%, 2 min 1,6-HD exhibited a greater impact (Additional file 1: Fig. S4C-G). Besides, parallel compartment strength analysis as previous study [40] also revealed a reproducible increase in the strength of compartmentalization upon 1.5%, 2 min 1,6-HD treatment (Additional file 1: Fig. S4H). In summary, short-term 1,6-HD treatment segregates compartments A and B (Fig. 3E).

We further examined genomic enrichment of biomolecular condensate components and key epigenomic features within strengthened or weakened compartment loci. We found that both weakened A and B loci exhibited slightly lower level of condensate-component binding (BRD4, MED1, RNAPII, HP1, and Ring1B) (Additional file 1: Fig. S4I, S4J) and key histone modification enrichment (H3K27ac, H3K4me1, H3K4me3, H3K27me3, H3K36me3, mC, and hmC) (Additional file 1: Fig. S4K, S4M). Besides, weakened A loci were of relatively lower PC1 value (Additional file 1: Fig. S4L).

## Short-term 1,6-HD treatment homogenized A-A compartment interactions

To examine the effects of 1,6-HD on interactions within compartment A or B, we visualized these interactions using saddle plots (Fig. 4A). Interestingly, the A-A interactions showed a "homogenized" pattern after 1–5 min (especially 2 min) of exposure to 1,6-HD. In the untreated cells, genomic loci with stronger A features (or high PC1 value) interacted more frequently with each other than those with weaker A features (or low PC1 value). Following short-term 1,6-HD treatment however, the interactions between all A-type loci, no matter strong or weak, became homogenized (Fig. 4A). We quantified this pattern by calculating average interactions between the loci in different PC1 ranges. Consistent with the saddle plots, interactions between stronger A features (high PC1 A, PC1 > 0.8) decreased and that between weaker A features (low PC1 A, 0.2<PC1 ≤ 0.4) increased (Fig. 4B-F, Additional file 1: Fig. S5A-B). We also examined the global "homogeneity" of A-A interactions at each time point and found that it was enhanced

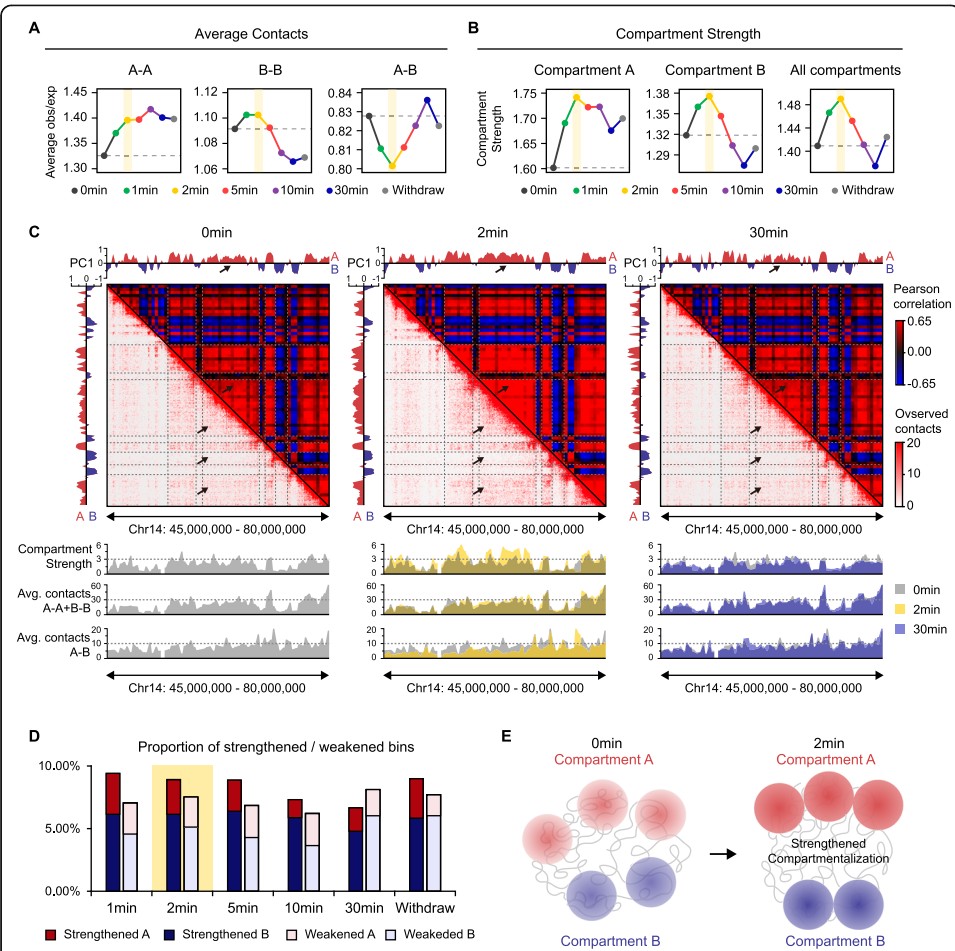

**Fig. 3** Short-term 1,6-HD treatment strengthened compartmentalization. **A** Average contacts between 100 kb bins from the same (A-A, B-B) and different (A-B) compartment types at each time point. The semi-transparent yellow shadow highlights the results at 2 min. **B** Compartment strength of all 100 kb bins from A-, B-, and both compartment types at each time. The semi-transparent yellow shadow highlights the results at 2 min. **C** Example (chr14: 45–80 Mb) showing compartmentalization strengthened at 2 min and weakened at 30 min. Top: eigenvector (PC1), red part refers to A compartment and blue part refers to B compartment. Middle: Pearson's correlation maps in the upper right and observed contact matrices in the bottom left. Bottom: per-bin compartment strength, per-bin average contacts between loci with same type (A-A or B-B), per-bin average contacts between loci with different types (A-B). **D** The proportion of strengthened (with at least 2-fold enhanced compartment strength) and weakened (with at least 2-fold reduction in compartment strength) bins compared with 0 min. **E** Schematic of the short-term 1,6-HD effects on compartmentalization

upon short-term treatment (Fig. 4C). These findings were further confirmed by chromosome conformation capture (3C)-qPCR assays (Fig. 4F). Taken together, the relative segregation of genomic loci with stronger A feature from the weaker ones can be disrupted by brief 1,6-HD treatment.

To understand the basic of this intra-A segregation, the gene functions, key epigenomic features, and genomic enrichment of biomolecular condensate components within the A-type loci in different PC1 ranges were analyzed. Remarkably, the regions with stronger A features (PC1 > 0.8) contained highly expressed genes enriched for basic cellular functions, and also had significantly more active histone modifications (H3K4me1, H3K27ac, H3K4me3, H3K36me3) (Fig. 4G, Additional file 1: S5A, S5B). In

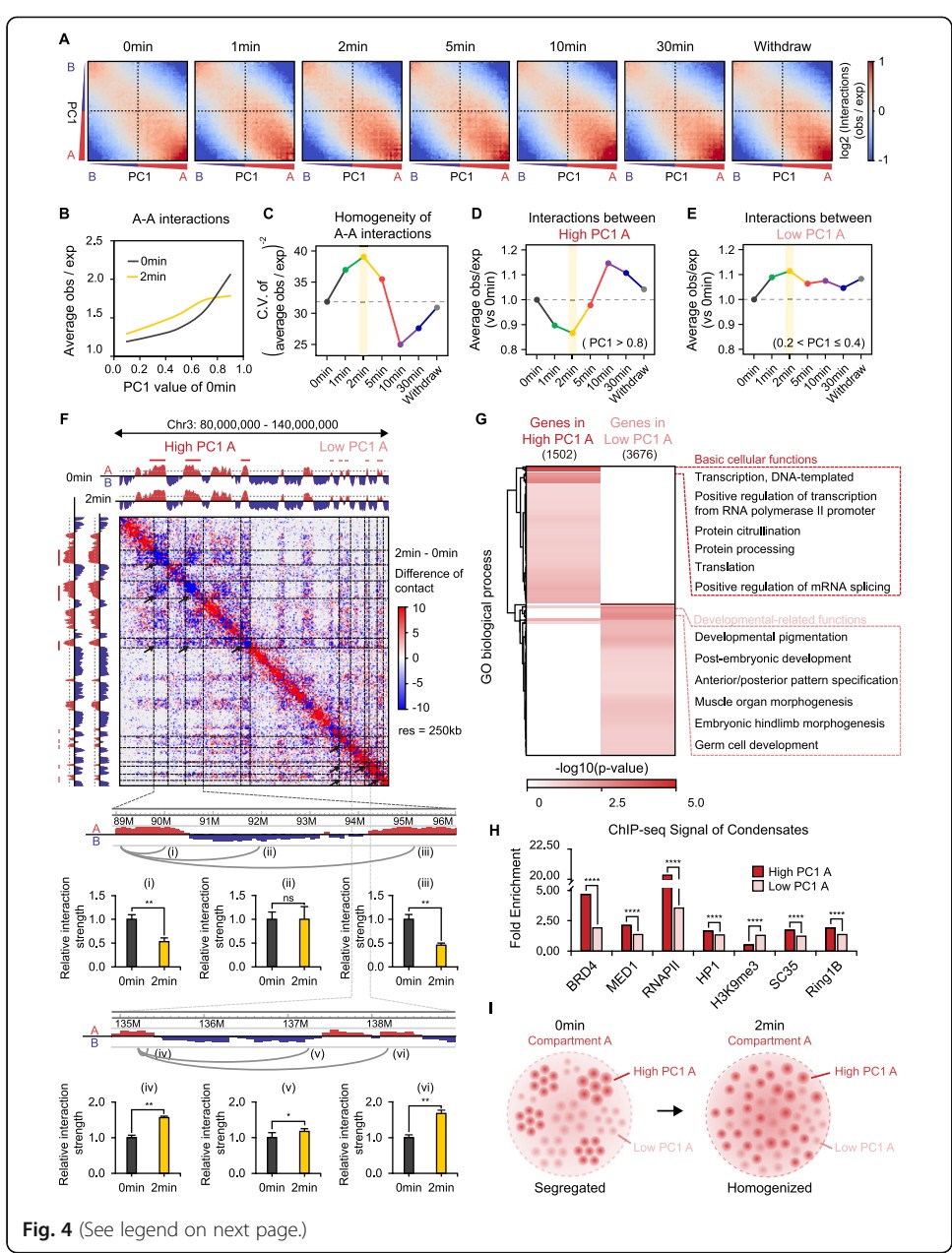

**Fig. 4** (See legend on next page.)

(See figure on previous page.)

**Fig. 4** Short-term 1,6-HD treatment homogenized A-A compartment interactions. **A S**addle plot at each time point indicating average contact enrichment between pairs of 500 kb loci arranged by their PC1 (shown on left and bottom). The lower right quarter represents A-A interactions, the upper left quarter represents B-B interactions while the upper right and lower left quarters represent A-B interactions. **B** Average A-A interactions of loci in different PC1 range. **C** Homogeneities of A-A interactions at each time point (details in "Materials and methods"). **D** Average interactions between strong A-featured loci (high PC1 A, PC1 > 0.8) versus 0 min at each time point. **E** Average interactions between weak A-featured loci (low PC1 A, 0.2 < PC1 ≤ 0.4) versus 0 min at each time point. **F** Top: Example (chr3: 80-140 Mb) showing homogenized A-A interactions at 2 min. Top left: PC1 indicates compartment assignments at 0 min and 2 min (A, red; B, blue). Center: heatmap showing difference of observed contacts between 2 min and 0 min (2 min minus 0 min). Bottom: fold change of strength of indicated interactions determined by 3C-qPCR assay. All *p* values were determined by Student's *t* test. Three biological replicates were assayed for 3C-qPCR experiment. **G** Clustering of gene oncology enrichment of genes in high PC1 A or low PC1 A. The blown box shows top 6 specific GO terms of each gene set. **H** Fold enrichment (median signal within interested regions / median signal across the whole genome) of ChIP-seq signal of condensate components. All *p* values were determined by the Wilcoxon rank-sum test. **I** Schematic indicating that short-term 1,6-HD exposure homogenized A-A interactions

contrast, the weaker A-featured regions (0.2<PC1 ≤ 0.4) contained genes enriched for development-related functions with relatively lower expression levels and had more negative histone modifications (H3K9me2, H3K9me3) (Fig. 4G, Additional file 1: S5C, S5D). Furthermore, the key components of biomolecular condensates were significantly more enriched in the stronger versus the weaker A regions (Fig. 4H). These results suggest a model wherein biomolecular condensates spatially segregate genes with different functions and activity in active compartments, and dissolution of these condensates disrupt functional segregation (Fig. 4I).

We also examined whether short-time 1,6-HD treatment had a similar "homogenizing" effect on B-type loci. Surprisingly, the B-type loci were largely unaffected by 2 min 1,6-HD treatment (Additional file 1: Fig. S5E-S5H).

## Short- and long-term 1,6-HD treatment respectively switched the intermediate compartments to A and B types

Despite widespread differences in compartmental interactions, the A/B compartment assignments were relatively consistent upon 1,6-HD treatment (Additional file 1: Fig. S6A). However, the duration of exposure affected PC1 clustering and compartment sizes (Fig. 5A, B, Additional file 1: Fig. S6B-C). While short-term 1,6-HD treatment significantly enlarged compartment A regions, long-time treatment multiplied the total length of compartment B regions (Fig. 5A, B). Consistent with this, 14.43% of the genomic loci experienced compartment switch due to 1,6-HD (Fig. 5C, Additional file 1: S6D). Most loci shifted to A upon short-term treatment and to B upon long-term treatment (Fig. 5C). In addition, majority of these "switch regions" were of low PC1 value (– 0.5 < PC1 < 0.5), indicating that they were relatively "intermediate" regions with weaker A- or B- features (Fig. 5D, Additional file 1: S6E). To further investigate the properties of the switch regions, we compared them with other low PC1 but constant regions in terms of PC1 distribution, size, key epigenetic features, and enrichment of biomolecular condensates. Consistent with the proportion of B-to-A and A-to-B switch during 1,6-HD treatment (Fig. 5C), the median PC1 of the switch regions was positive and increased upon short-term treatment before shifting to negative with longer exposure. In contrast, the median PC1 of constant regions remained negative during the entire

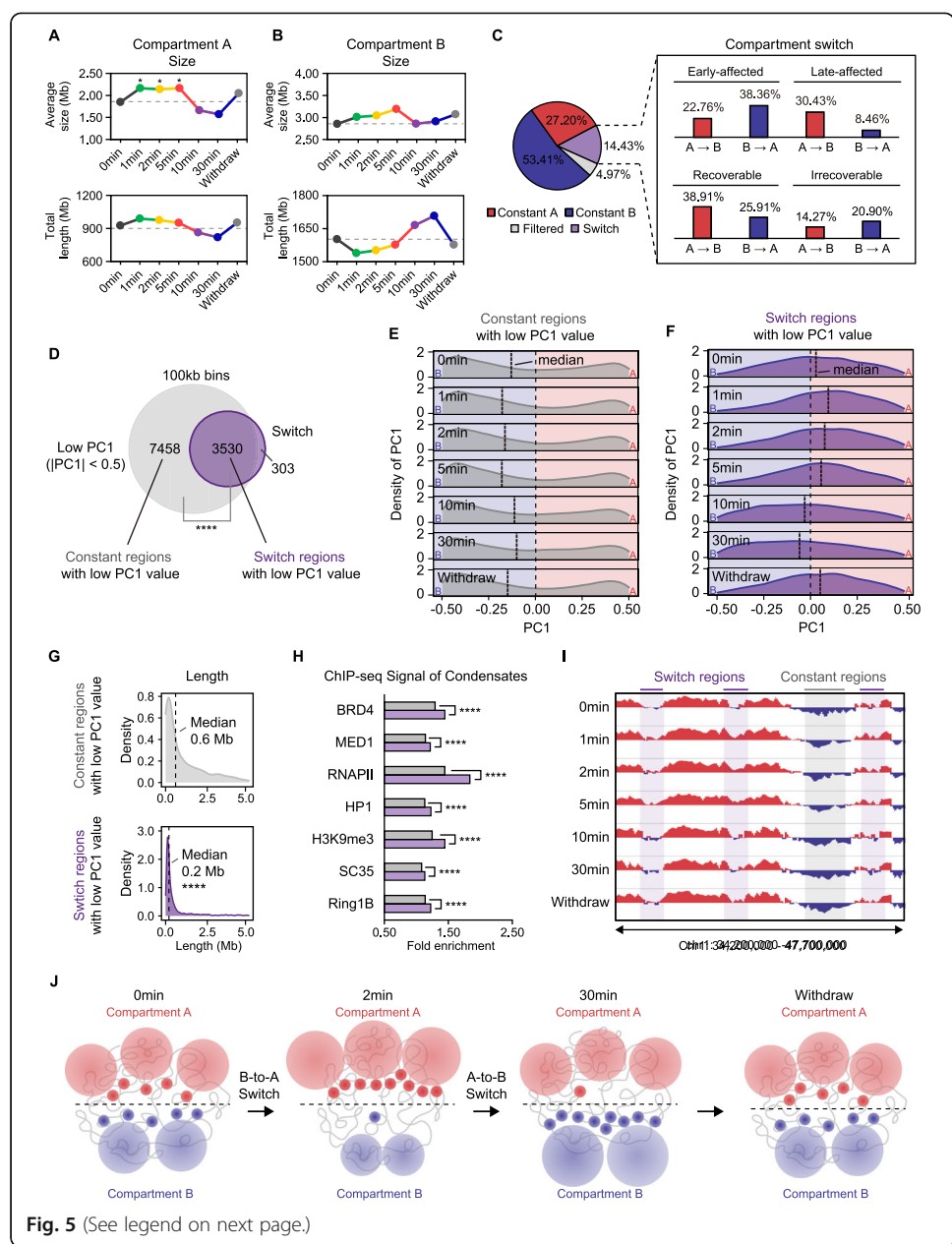

**Fig. 5** (See legend on next page.)

(See figure on previous page.)

**Fig. 5** Short and long-term 1,6-HD treatment respectively switched the intermediate compartments to A and B types. **A** Top: average size of compartment A regions. Bottom: total length of compartment A regions. All *p* values were determined by the Wilcoxon rank-sum test. **B** Top: average size of compartment B regions. Bottom: total length of compartment B regions. All *p* values were determined by the Wilcoxon rank-sum test. **C** Proportion of constant A, constant B, switch and filtered regions (with low mapping ration or high copy number). The blown box showing the details of switch regions. "Early-affected" refers to switch regions in 1/2/5 min while "Late-affected" refers to switch regions in 10/30 min. "Recoverable" refers to switch regions have same state in withdrawal and 0 min while "Irrecoverable" refers to switch regions have different states in withdrawal and 0 min. **D** Venn plot of switch regions (purple) and regions with low PC1 value (− 0.5 < PC1 < 0.5) (grey). The *p* value was determined by hypergeometric test. **E** Density plot of the PC1 distribution of constant regions with low PC1 value at each time point. Black dashed line indicates the median value. Background color indicates compartment assignments (A, red; B, blue). **F** Density plot of the PC1 distribution of switch regions with low PC1 value at each time point. Black dashed line indicates the median value. Background color indicates compartment assignments (A, red; B, blue). **G** Density plot of size distribution of constant regions and switch regions with low PC1 value. Black dashed line indicates the median value. The *p* value was determined by the Wilcoxon rank-sum test. **H** Fold enrichment (median signal within interested regions / median signal across the whole genome) of ChIP-seq signal of condensate components. All *p* values were determined by the Wilcoxon rank-sum test. **I** Example (chr1: 34.2–47.7 Mb) showing constant regions and switch regions with low PC1 value. Each track: PC1 indicates compartment assignments at each time point (A, red; B, blue). **J** Schematic indicating that short- (2 min) and long-term (30 min) 1,6-HD treatment respectively switched the intermediate compartments to A and B types

treatment duration (Fig. 5E, F). The switch regions were notably distinct from the constant regions in terms of their smaller sizes and significantly higher enrichment of key histone modifications (Fig. 5G, Additional file 1: S6F, S6G). Furthermore, the switch regions had significantly more enrichment of key components of biomolecular condensates compared to the constant regions (Fig. 5H). In summary, the small intermediate compartments with relatively more binding of condensate factors are sensitive to 1,6-HD, and switch to A or B depending on the treatment duration (Fig. 5I, J).

## Short-term 1,6-HD treatment respectively enlarged and shortened the TADs in A and B compartments

Given the differential response of the A and B compartments to 1,6-HD, we next analyzed their respective TADs. TADs were identified by TopDom [41], a tool known to be robust to resolution and sequence depth, and the identification of TADs was verified by comparison to the contact maps from ultra-deep in situ Hi-C data and the high density of CTCF peaks on TAD boundaries (Additional file 1: Fig. S7A, S7B). As shown in Fig. 6A and Fig. S7C, the TADs in compartment A were enlarged upon 1,6-HD treatment and did not recover after withdrawal. In contrast, the TADs in compartment B were respectively shortened and lengthened by short-term and long-term 1,6-HD treatment, and their size recovered after withdrawal (Fig. 6B, Additional file 1: Fig. S7D). Consist changes of the length of chromatin loops in compartment A or B were also observed upon 2 min treatment (Additional file 1: Fig. S7E-F). The TADs with significantly altered boundaries relative to the untreated cells were classified into fusion, shifting, and separation. Corresponding to the changes in size, TADs in A compartment underwent more fusion and shifting than separation during 1,6-HD exposure, whereas a higher proportion of TADs in B compartments exhibited separation following short-term versus long-term treatment (Fig. 6A, B). These findings were confirmed by 3C-qPCR assays, where interactions between fused TADs in A compartment

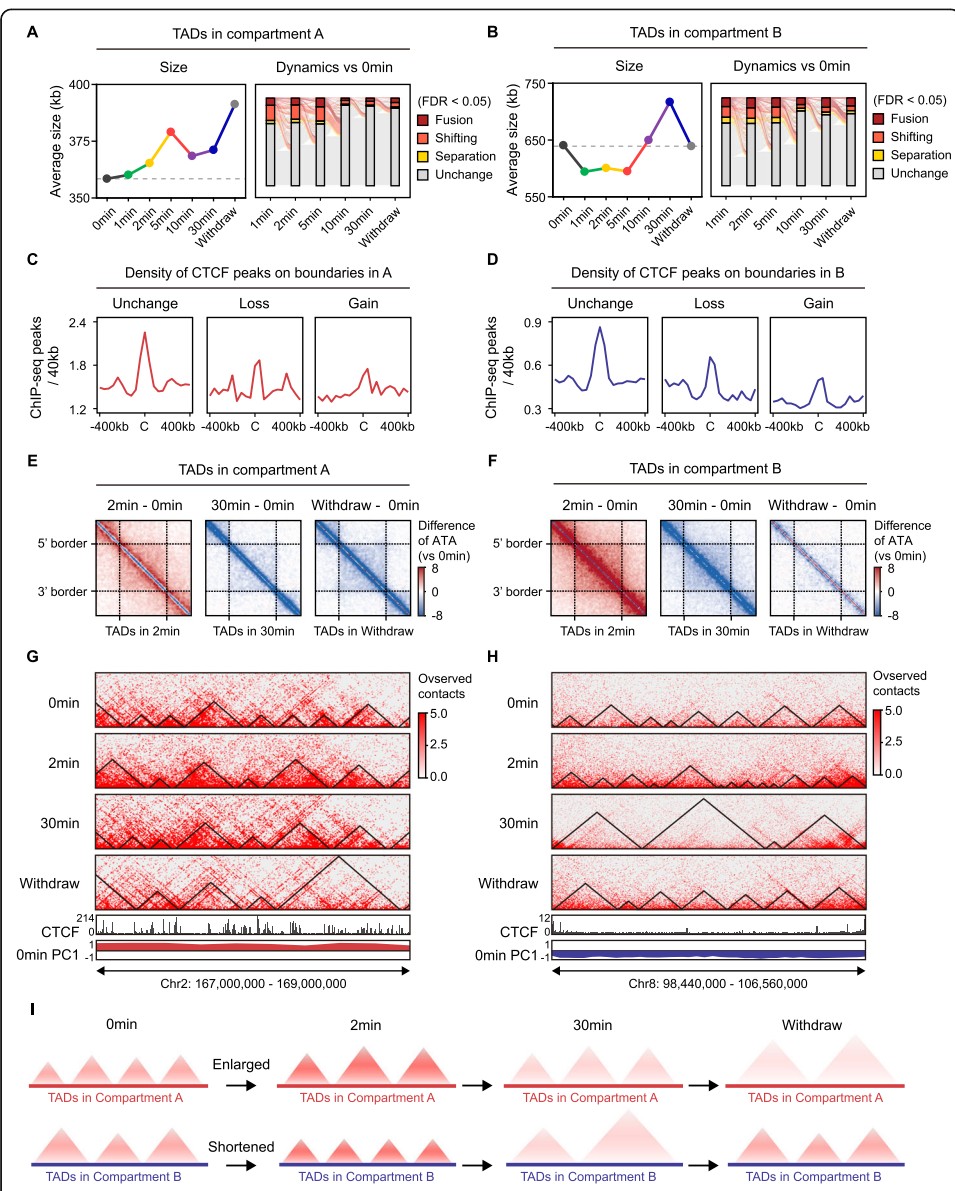

**Fig. 6** Short-term 1,6-HD treatment respectively enlarged and shortened the TADs in A and B compartments. **A, B** Left: average size of TADs in compartment A (**A**) or compartment B (**B**). Right: alluvial plot of TAD reorganization in compartment A (**A**) or compartment B (**B**) during 1,6-HD treatment. Line thickness is proportional to the number of TADs with a specific type of reorganization compared to 0 min. All FDR values were determined by diffHic (details in "Materials and methods"). **C, D** Density of CTCF ChIP-seq peaks in unchanged, lost, or gained boundaries in compartment A (**C**) or compartment B (**D**) across all time points compared to 0 min. **E, F** Difference of aggregate TAD analysis (ATA) signal versus 0 min. TADs in compartment A (**E**) or compartment B (**F**) are resized to a uniform size and calculated the average interactions. **G, H** Examples (**G**: chr2:167–169 Mb, **H**: chr8: 98.44–106.56 Mb) showing TADs in compartment A (**G**) or compartment B (**H**) at 0 min, 2 min, 30 min, and withdrawal. Bottom: CTCF ChIP-seq signal and PC1. **I** Schematic indicating that short-term 1,6-HD respectively enlarged and shortened the TADs in A and B compartments

significantly increased and interactions within separated TADs in B compartment significantly decreased (Additional file 1: Fig. S7G-H). In addition, TADs in both compartments were significantly more dynamic in response to short-term (~ 30% in A and B) as opposed to long-term (~ 10% in A, ~ 15% in B) treatment (Fig. 6A, B). Furthermore, both the lost and gained boundaries had a lower peak density of the structuring factor CTCF compared to the unaltered boundaries (Fig. 6C, D). Besides, lost boundaries in both A and B exhibited significantly lower level of expression (Additional file 1: Fig. S7I-L). These results imply that biomolecular condensates may play a role in stabilizing TAD boundaries with less CTCF density.

To assess whether the intra-TAD interactions were affected by 1,6-HD treatment, we performed aggregate TAD analysis (ATA) to calculate average interactions within TAD. The intra-TAD interactions were enhanced upon 2 min treatment and reduced after 30 min, which was consistent with the changes in short-range interactions at each time point (Figs. 2B, 6E-H). In addition, the reduction in intra-TAD interactions in the A compartment did not recover after 1,6-HD withdrawal, whereas the TADs in B largely recovered (Fig. 6E-H). Taken together, short-term 1,6-HD treatment led to TAD reorganization and compaction, with enlargement and shortening of the TADs in compartments A and B respectively (Fig. 6I).

## Interactions between condensate-enriched regions were long-range and weakened upon 1,6-HD treatment

Previous results have shown that 3D genome reorganization in response to 1,6-HD is related to the genomic binding of key components of biomolecular condensates, we next sought to determine whether the interactions between specific condensate-component enriched regions were affected by 1,6-HD treatment. Since the long-range interactions decreased significantly upon 1,6-HD treatment (Fig. 2A, B), we hypothesized that the interactions between the condensate-component-enriched regions are long-range. To test this, we used a statistical model to determine significant interactions between all TADs in untreated cells and identified "condensate-enriched interactions" wherein bins at both ends overlapped with top 10% ChIP-seq signals of key components of specific condensates. As a control, we also identified the interactions between bins overlapping with bottom 10% signals as "condensate-absent interactions," as well as the interactions between randomly selected 10% bins as "random interactions." In agreement with our hypothesis, the condensate-enriched interactions were significantly longer (Fig. 7A). Furthermore, aggregate peak analysis (APA) was used to observe the aggregated signals of condensate-enriched interactions during the treatment duration to determine whether these long-range condensate-enriched interactions were affected by 1,6-HD (Fig. 7B). Consistent in all type of condensates we examined, the condensate-enriched interactions were initially strong but weakened considerably after 1,6-HD treatment, and slightly recovered after withdrawal (Fig. 7B, C). We also examined the effects of 1,6-HD on structural factor-mediated interactions, including CTCF and SMC1A, which were respectively identified by ChIA-PET and HiChIP experiments in previous studies [24, 42]. Different with condensate-component-enriched interactions, these structural factor-mediated interactions remained mainly unaffected (Fig. 7D, E). Taken together, the condensate-enriched interactions are long-range and

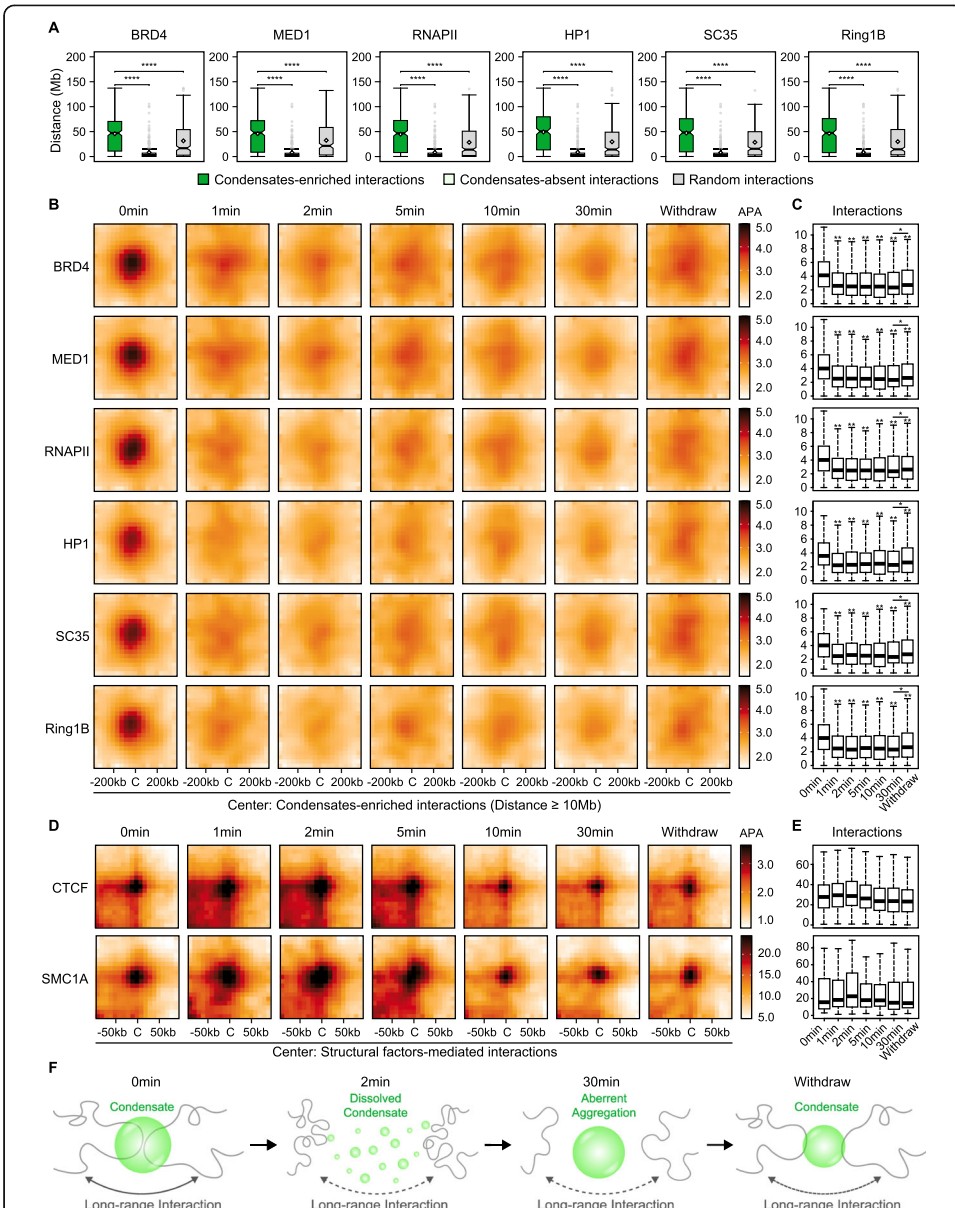

**Fig. 7** Interactions between condensate-enriched regions were long-range and weakened upon 1,6-HD treatment. **A** Distance of "condensate-enriched interactions," "condensate-absent interactions," and "random interactions." Condensate-enriched interactions: both ends overlapping with 40 kb bins with top 10% ChIP-seq signals of key components of specific condensates. Condensate-absent interactions: both ends overlapping 40 kb bins with bottom 10% signals. Random interactions: both ends overlapping randomly selected 10% bins. **B** Aggregate peak analysis (APA) of long-range condensate-enriched interactions (≥ 10 Mb) at 40 kb resolution. **C** Average interactions surrounding the center (< 120 kb) of long-range condensate-enriched interactions (≥ 10 Mb). All *p* values were determined by the Wilcoxon rank-sum test. **D A**ggregate peak analysis (APA) of structural factor-mediated interactions at 10 kb resolution. **E** Average interactions surrounding the center (< 30 kb) of structural factor-mediated interactions. Pairwise Wilcoxon rank-sum test was performed but no significant changes were detected. **F** Schematic indicating that interactions between condensate-enriched regions are long-range and weakened upon 1,6-HD treatment

weakened by 1,6-HD, which may account for the global 3D chromatin reorganization at different hierarchies (Fig. 7F).

## Discussion

We found that 1.5%, 2 min is a proper condition of 1,6-HD usage to dissolve condensates with minimal side effects in mESCs. Since the basic biochemical mechanism of protein condensation is common across cell types [28, 43–46], this standardized 1,6-HD condition may help analyze dissolution of condensates in other cell types as well, and in fact did show encouraging results in mouse embryonic fibroblasts (MEFs) and neural progenitor cells (NPCs) (Additional file 1: Fig. S2). Besides, we also observed a dissolution of paraspeckles upon 1.5%, 2 min 1,6-HD by live-cell imaging in HeLa cells (Additional file 1: Fig. S1C). Therefore, when applying 1,6-HD treatment to other cell types, this condition would be a valuable reference.

Notably, the two widely used concentrations of 1,6-HD (5% and 10%) [3, 9, 11, 47–51] resulted in massive cell death of mESCs even in short time (2 min) (Additional file 1: Fig. S1A). These conditions seem to work well in previous studies, probably owing to the fact that some studies only focused on dissolving specific type of condensate they are interested in without detecting cell variability, or used tumor-derived cell lines (e.g., Hela [49], K562 [50]) which show activation of anti-apoptotic pathways [52]. Despite cell variability, cell size also matters. Cell shrinkage and increased macromolecular crowding are often accompanied by aberrant protein aggregations [9]. In addition, 5% or higher concentration of 1,6-HD drastically and irreparably suppressed chromatin motion and hyper-condensed chromatin in live cell [13]. These side effects of unproper 1,6-HD using conditions on cell viability, cell size, and chromatin behavior become particularly problematic when exploring global role of condensates using large number of cells. We therefore recommend relatively moderate condition (low concentration, short time) and careful test of cell variability and cell size to ensure that 1,6-HD is working properly. Overall, optimal condition of 1,6-HD usage is worthwhile for researchers to explore the global relationship between condensates and other interested biological process or structure in living cells.

Two recent studies treating cells with high concentration (5% and 10%, respectively) and long time (15 min and 20 min, respectively) of 1,6-HD have reported different results in 3D reorganization [49, 50]. One observed moderate 3D reorganization upon treatment [49] whereas the other observed a global 3D reorganization that short-range interactions decreased and long-range interactions increased [50], which is consistent with our observation in cells upon 1.5%, long-term (10 min and 30 min) 1,6-HD treatment and opposite to our results in proper condition. The opposite effect of different conditions of 1,6-HD treatment might occur because high concentration (5% or 10%) of 1,6-HD treatment caused chromatin "freeze [13]," chromatin hyper-condensation, nucleus shrinkage, and abnormal protein aggregates in live cells, while low condition and short-time treatment does not have those side effects (Additional file 1: Fig. S1). These phenotypes are likely caused by a loss of membrane function and a potential loss of water from the cells that leads to an increase in macromolecular crowding, since the mechanism of 1,6-HD is to dissolve condensates by disrupting hydrophobicity.

The process of phase separation to form condensates has recently been proposed to have a prominent role in genome compartmentalization [11, 19, 21, 39]. Interactions

between heterochromatic regions have been reported to be essential for spatial segregation of the active and inactive compartments [39], while the heterochromatin domain formation has been reported to be driven by phase separation of HP1 [11, 18, 19]. Therefore, genome compartmentalization is expected to be weakened upon condensate-dissolving. Interestingly, however, global compartmentalization is strengthened by 1.5%, 2 min 1,6-HD treatment. This is probably because a proportion of (hetero)chromatin is of high density and exhibits solid-like properties, which is hard to be affected by 1.6-HD [11]. Upon treatment, this part of chromatin may become more distinctly segregated from other mobile regions, and thus result in the strengthened compartmentalization. Besides, not only HP1, but many other condensates are also disrupted by 1,6-HD. Some of these condensates may contribute to maintain A-B interactions. Nevertheless, despite the A/B segregation, our results strongly propose that biomolecular condensates help the spatial segregation of genes with different functions and transcriptional levels in active compartments. Consistent with this, a previous study showed that disruption of nuclear speckles by *Srrm2* depletion significantly decreased the top 20% A-A interactions [53]. These findings provide an experimental evidence for a simple and intuitive mechanism that can explain the tendency of the genome to compartmentalize similar biochemical process, and thus link the biomolecular condensates and 3D genome to cellular functions.

To understand the mechanisms underlying the changes in chromatin structures, we directly investigated interactions between condensate-component-enriched regions and found that they were long-range and weakened upon 1,6-HD treatment. Previous studies also suggest a close relationship between long-range chromatin interactions and biomolecular condensates [25, 54]. For example, the SPRITE approach that can identify interactions across larger distances indicated that long-range interactions are organized around nucleolus and nuclear speckles [25]. Taken together, our findings suggest a model wherein biomolecular condensates mediate specific long-range interactions, and their disruption may result in global disorders of high-order chromatin structures.

Previous studies observed that degrading BRD4 or mediators have little impact on enhancer-promoter interactions (E-P interactions) [35, 55], which are mainly mediated by cohesin. There are several differences between condensate-component-enriched interactions we focused on in this study and E-P interactions. Compared with the E-P interactions, the condensate-component-enriched interactions across larger genomic distances (Additional file 1: Fig. S8A) and with lower cohesin peak density at anchors (Additional file 1: Fig. S8B). In fact, E-P interactions also remained unaffected upon 1,6-HD treatment (Additional file 1: Fig. S8C). Furthermore, El Khattabi et al. [55] speculates that the underlying proteins which drive E-P interactions may display a great deal of redundancy, because depletion of RNAPII and mediator does not overly disrupt chromatin topology. The redundancy here means that the protein "hubs" containing different types of components may still exist when one particular component was depleted. To preliminarily test this point, we degron BRD4 using AT1 [35] and identified the condensation of RNAPII, which also plays role in chromatin loop maintenance [31]. As a result, RNAPII condensates remained unaffected in the absence of BRD4 (Additional file 1: Fig. S8D). These results imply that the redundancy of condensate components may exist.

Since 1,6-HD is a tool to globally disrupt condensates, it is hard to rule out whether the disruption of condensates in cytoplasm or on cell membrane after 1,6-HD treatment has an impact on 3D chromatin organization. Nevertheless, it still makes sense to

study the change of 3D chromatin organization upon 1,6-HD treatment. We did find chromatin organization changes are related to neighboring nuclear condensates through integrated analysis of ChIP-seq data of components of condensates and the interactions between the condensate-component-enriched regions were greatly affected by 1,6-HD treatment. This global view of 3D chromatin reorganization after dissolving condensates should help to figure out possible models of the detailed molecular mechanisms and provide clues for further investigation of particular nuclear condensate.

Emerging evidences show that aberrant phase transitions of proteins contribute to a large set of neurodegenerative diseases [56]. For instance, the liquid-to-solid phase transition of FUS protein is related to amyotrophic lateral sclerosis (ALS) [57, 58]. In this study, we provided a time-resolved map of 3D genome during a 1,6-HD-induced phase transition process of biomolecular condensates, which illustrated the 3D genome disorders accompanied by aberrant phase transitions. This may help researchers to further investigate the mechanism of phase-related diseases in the aspect of 3D genome and may shed light on some plausible novel therapeutic routes to diseases.

## Conclusions

In conclusion, our study finds a proper 1,6-HD condition (1.5%, 2 min) to dissolve biomolecular condensates with limited side effects and provides a time-resolved map of chromatin organization during 1,6-HD treatment. We demonstrate that 1,6-HD treatment has a time-dependent effects on biomolecular condensates and chromatin organization at different hierarchies. This time-resolved map would serve as a resource for exploring the role of biomolecular condensates in 3D chromatin organization.

## Materials and methods

### Cell culture and differentiation

R1 mESCs (a kind gift from Dr. Shaorong Gao [59]) were grown on gelatin-coated dish with ESC medium, which consisted of DMEM (Hyclone), 15% (v/v) fetal bovine serum (FBS; Thermo Fisher), 0.1 mM β-mercaptoethanol (Sigma), 2 mM L-glutamine (Thermo Fisher), 0.1 mM nonessential amino acids (Thermo Fisher), 1% (v/v) nucleoside mix (Sigma), and 1000 U/mL recombinant LIF (Millipore). MEFs, mRuby3-NONO-KI HeLa cells [60], and H2B-GFP stably expressed HeLa Cells were cultured in DMEM (Hyclone) with 15% (v/v) fetal bovine serum (FBS; Thermo Fisher). NPCs were grown in neural basal (Gibco) supplemented with 1× NEAA (Gibco), 0.5 × GlutaMAX (Gibco), 1 × B27 (Gibco), 1 × N2 (Gibco), 0.075% BSA (Sigma), 1 × PS (Hyclone), 10 ng/mL bFGF (PeproTech), 10 ng/mL mEGF (PeproTech).

To induce NPC differentiation [61], monolayer culture for neural differentiation was performed. Briefly, mESCs were dissociated and plated at a density of $1 \times 10^4$ cells/cm$^2$ in N2B27 medium (DMEM/F12, Neurobasal, N2 (Thermo Fisher 17502-048), B27 (Gibco 17504044)) supplemented with 1 mM L-glutamine and 0.1 mM β-mercaptoethanol.

The MEFs isolated from E13.5 to 14.5 days of pregnancy could be maintained in culture consistently for more than 10 passages, and the third or fourth passage was used for experiments.

The cell lines have been authenticated and are available upon request.

## Chemical treatment of cells

The ESC medium contained different concentrations of 1,6-HD (Sigma, 240117) or 2,5-HD (Sigma, H11904) were prepared and warmed in 37 °C incubator. For treating, the medium in the dish is discarded, and the homogeneous and warmed medium contained different concentrations of 1,6-HD or 2,5-HD were added. Then, the dish was put into the incubator immediately. Treated cells were immediately used for corresponding experiments. BRD4 degradation was performed as previously described [35], and the mESCs were cultured in ESC medium contained 1 μM AT1(Tocris, 6356/5) for 24 h.

## Apoptosis

The apoptosis of cells was measured using eBioscience™ Annexin V Apoptosis Detection Kit APC, according to the manufacturer's instructions

## Immunofluorescence (IF)

For immunofluorescence, mESCs were grown on gelatin-coated glass for 24 h. Fresh media with different concentrations of 1,6-HD was added on the plate in different time length. Treated cells were immediately fixed in 4% paraformaldehyde (PFA) for 10 min. After washes for three times, cells were permeabilized with 0.25% Triton X-100. Following washes in PBS, cells were blocked with 10% bovine serum albumin (BSA, Sigma) for 1 h at room temperature. Then cells were incubated with primary antibodies, including BRD4 (Abcam, ab128874), MED1 (Abcam, ab64965), HP1a (Abcam, ab109028), RING1B (CST, 5694), SC35 (Abcam, ab11826), CTCF (Santa Cruz, sc-271474), SMC1A (Santa Cruz, sc-393171), SMC1A (Bethyl laboratories, A300-055A), Pol II (ABclonal, A2107), H3K9me3 (Abcam, ab8898), and anti-Nucleolin (Abcam, ab22758) in 5% BSA overnight at 2–8 °C. After washes in PBS, cells were incubated with Alexa 488-labeled anti-rabbit (Thermo Fisher) or Green Donkey Anti-Mouse (IFKine) secondary antibodies for 1 h. Then, cells were washed three times in PBS. Cells were incubated with DAPI in PBS. The fixed cells were imaged by using Nikon Structured illumination microscopy (N-SIM) and analyzed using NIS-Elements or ImageJ with same parameter across different groups. Nuclear volumes were determined from DAPI staining through the z-stack image, which was then processed through NIS-Elements. For each cell type under different conditions, 30 cells were used for quantification of nuclear volume.

## Puncta analysis

The fixed cells after immunofluorescence were imaged by N-SIM with same parameter across different groups. The puncta number inside the nucleus was calculated by the spot module of imaris software (Bitplane). The Slice module was used to measure diameter of puncta. The measured diameter of puncta was input to Surpass module as parameters to calculate the puncta number in selected cell. Furthermore, the Spots1 module in Surpass was used to select the "Region of Interest" for calculating the puncta number in nucleus (DAPI-stained regions). For each type of condensate under different conditions, 30 cells were used for quantification of puncta.

## DNA-FISH combined with IF

Cells were fixed with 4% PFA for 12 min, washed in PBS for twice. After removing background with 1 mg/mL NaBH4, cells were permeated with 1% Triton X-100 for 10

min at room temperature, followed by incubation at 37 °C in 100 μg/mL RNaseA for 45 min to remove RNA. Then cells were incubated with anti-MED1 antibody at 4 °C overnight, followed by incubated with secondary antibody for 1 h at room temperature. Cells were treated with 50% formamide in 2 × SSCT at 4 °C overnight to loosen the chromatin, followed by heating at 78 °C for 10 min, and incubated in 70%, 85%, and 100% ethanol for 1 min, respectively. Primary probe in hybridized buffer with 50% formamide, 10% dextran sulfate, 1% Triton X-100, and 2 × SSC was added into dish and incubated with cells 37 °C for 20 h. After washing 15 min in 2 × SSC twice, cells were incubated with secondary probe in 2 × SSC with 30% formamide for 30 min. After washing with 2 × SSC for 3 times, cells were imaged with a microscope.

The fluorescent DNA hybridization probes were designed and generated using ULYSIS® Nucleic Acid Labeling Kits (Thermo, U21660 and U21654). *Nanog SE*, design Region: (mm9) Chr6: 122605985-122705985. *Pou5f1 SE*, design Region: (mm9) Chr17: 35617900-35690000.

### Live-cell imaging

For live-cell imaging of paraspeckles, mRuby3-NONO-KI HeLa cell lines [60] were cultured in dishes (PerkinElmer, 6055300) and observed in DMEM (Hyclone) by Lionheart FX Automated Microscope (BioTek). A humidity chamber and $CO_2/O_2$ Gas Controller (BioTek) were used to maintain cell culture condition (37 °C, 5% $CO_2$ and humidity). Live cells fluorescently labeled with mRuby3 were excited by the 561-nm laser through an objective lens (× 40) and detected at 647 nm. Images were automatically collected at 1 min/frame. For super-resolution live-cell imaging of H2B-GFP, H2B-GFP stably expressed HeLa cells were cultured in 35-mm glass bottom dishes (Cellvis). The live-cell was cultured in a stage-top incubator (INUG2H-TIZSH-SET, Tokai Hit, Japan), and cell culture conditions (37 °C, 5% CO2, and humidity) during imaging were maintained. Single nucleosomes were observed by an inverted microscope (ECLIPSE Ti-E with N-STORM module, Nikon, Japan) and a sCMOS camera (ORCA-Flash 4.0, Hamamatsu Photonics, Japan). Live cells labeled with H2B-GFP were excited by a 488-nm laser and detected at 500–550 nm. An oblique illumination system (the TIRF function of the N-STORM module) was used to excite the fluorophore within a limited thin area in the cell nucleus and reduce background noise. Sequential image frames were acquired using NIS-Elements software (Nikon) at a frame rate of 50 ms under continuous illumination.

### Trajectory analysis

Sequential images were converted to 8-bit grayscale, and the background noise signals were subtracted with the rolling ball background subtraction (radius, 50 pixels) of ImageJ (v1.53j, https://imagej.net/Fiji). The centroid of each fluorescent dot and its trajectory was tracked with Fiji plugin TrackMate [62] using the LoG detector. Mean square displacement (MSD) shows the spatial extant of dot motion in a certain time period [63] was calculated using the data from TrakeMate by a published Python program [64], which multiplied the 2D MSD by 1.5 to get the 3D value.

### Clustering analysis of nucleosomes

Super-resolution images were converted to 8-bit grayscale, and the background noise signals were subtracted with the rolling ball background subtraction (radius, 50 pixels)

of ImageJ (v1.53j, https://imagej.net/Fiji). The methods for clustering analysis of nucleosomes were same as previous study [65]. Ripley's K function can be calculated by the average particle density of area S:

$$K(r) = \left(\frac{S}{N-1}\right)\left[\frac{1}{N}\sum_{i=1}^{N}\sum_{i\neq j}\delta(r-r_{i,j})\right]$$

where $N$ is the total number of dots in the area S. The $\delta$ function is given by

$$\delta(r-r_{i,j}) = \begin{cases} 1, & r_{i,j} \leq r \\ 0, & r_{i,j} > r \end{cases}$$

where $r_{i,j}$ is the distance between $r_i$ and $r_j$, and the $L$ function is given by

$$L(r) = \sqrt{\frac{K(r)}{\pi}}.$$

The area $S$ of the total nuclear region was estimated using the Fiji plugin Trainable Weka Segmentation, and the area of the whole region was measured by Analyze Particles (ImageJ).

### BAT-Hi-C

BAT-Hi-C was performed according to a previously published protocol [66]. In brief, cells were treated with ESCs media with 1.5% 1.6-HD or ESCs media only for different time length. One percent formaldehyde in PBS was used for crosslinking of cells for 10 min, followed by quenching with glycine at a final concentration of 125 mM. Cells were washed with cold PBS and harvested by scraping cells in cold PBS with protease inhibitor. Collected cells were pelleted at 1350$g$ for 5 min at 4 °C. Genome was digested with AluI, and the ends of restriction fragments were labeled using biotinylated nucleotides. After reversal of crosslinks, ligated DNA was purified and sheared to a length of 300–500 bp. After purification, DNA was prepared for sequencing. Each group was prepared with two replicates.

### Chromatin immunoprecipitation (ChIP)

ChIP was performed as previously described [67]. Briefly, $5 \times 10^7$–$1 \times 10^8$ mESCs were crosslinked with 1% formaldehyde for 10 min at room temperature, followed by quenching with 125 mM glycine. Rinse cells twice with ice-cold PBS and then resuspended in lysis buffer with protease inhibitor. Chromatin was sonicated into 200–500 bp range. Dynabeads (Invitrogen 10004D) pre-blocked with 0.5% BSA were incubated with 10 μg antibody (BRD4 (Abcam, ab128874), MED1 (Abcam, ab64965), CTCF (ABclonal, A1133), SMC1A (Bethyl laboratories, A300-055A)) for more than 8 h. Chromatin was added to antibody-bead complex and incubated rotating overnight at 4 °C. Beads were washed with each of the following buffers: Wash Buffer (RIPA), TE + 50 mM NaCl at 4 °C, followed by washing one time with TE at room temperature. Chromatin was eluted by incubating at 65 °C for 45 min in elution buffer. Reversal of crosslinking was performed at 65 °C overnight. After treatment with 10 mg/ml RNaseA and 20 mg/ml proteinase K, DNA fragments were then purified using phenol–chloroform extraction and resuspended in 10 mM Tris-HCL.

### 3C-qPCR assay

3C was performed as previously described [68]. Cells were crosslinked with 1% formaldehyde for 10 min at room temperature. Cells were resuspended in 500 μl of lysis buffer (10 mM Tris-HCl pH 8.0, 10 mM NaCl, 0.2%Igepal CA630, protease inhibitors cocktail) and incubated for 15 min at 4 °C. Cells were washed twice in lysis buffer and incubated in 50 μl of 0.5% SDS at 65 °C for 8 min. SDS was quenched by incubating in a solution of 25 μl of 10% Triton X and 145 μl $H_2O$ at 37 °C for 15 min. Then, 25 μl of 10 × NEBuffer2 and 20 μl of AluI were added and the chromatin was digested at 37 °C overnight with rotation. To inactivate AluI, the sample was incubated at 65 °C for 20 min. The restricted fragments were added into a solution of 663 μl $H_2O$, 120 μl NEB T4 ligase buffer, 100 μl 10% Triton X-100, 12 μl 10 mg/ml BSA, and 5 μl T4 DNA ligase. The ligation was performed for 4 h at room temperature with rotation. Crosslinks were reversed by adding 50 μl of 20 mg/ml proteinase K, 120 μl 10% SDS at 55 °C for 30 min, and another addition of 130 μl 5 M NaCl at 68 °C overnight. The DNA was precipitated with ethanol and resuspended in $H_2O$ for about 100 ng/μl DNA for qPCR. For qPCR experiments, qPCR was performed using SYBR qPCR Master mix (Vazyme) and the primers used are listed in Additional file 2. Neighboring primers at the *B2m* gene locus reported in a previous study were used for normalization [69].

### Hi-C analysis

#### Data processing

Adapters were first trimmed with Cutadapt (version 2.4, http://code.google.com/p/cutadapt). Next, bridge linkers of paired-end reads were trimmed with trimLinker program (part of ChIA-PET2, version 0.9.3, https://github.com/GuipengLi/ChIA-PET2, parameters: "-t 20 -m 1 -k 1"). Subsequently, the Hi-C paired-end reads were aligned to the mm9 reference genome and paired using HiC-Pro [70] (version 2.11.1, parameters: default settings). To normalize the data, valid pairs were randomly sampled to the smallest given read number across all replicates. The correlation of Hi-C matrices was analyzed by hicCorrelate (part of hicexplorer [71], version 3.5.3, parameter: "--range 5000:200000, --method = pearson, --log1p"). The two replicates were merged in the following analysis and visualization (except for differential domain boundaries detection, details in "TAD analysis"). Then the data was used to generate contact matrices and corrected with ice_norm [72] (part of HiC-Pro). For each chromosome, the ICE-normalized 40 kb, 100 kb, and 500 kb resolution contact matrices were used for further analysis.

#### Relative contact probability

The 40 kb ICE-normalized contact matrices of each chromosome were used to analyze relative contact probability (RCP) and the RCP log2 foldchange to 0 min with GENOVA (version 0.9, https://github.com/robinweide/GENOVA).

#### Quantification of changed interactions at certain genomic distance

Pair of bins in the ICE-normalized contact matrices across certain genomic distance was extracted and compared with the mean contact frequency in the same bin pair of the two replicates in 0 min. For different genomic distance, matrices at different

resolution were used. In detail, 500 kb matrices were used for quantification of 1 Mb distance, 100 kb matrices were used for 10 Mb distance, 40 kb matrices were used for 1 Mb distance, and 25 kb matrices were used for 100 kb distance.

### Compartmentalization

For compartmentalization analysis, the first eigenvector (PC1) values were calculated from ICE-normalized matrices (100 kb bin) at each chromosome separately, with CscoreTool [73] (version 1.1, parameters: 20 1000000). Next, each bin was assigned into A or B compartments according to its PC1 values. The gene-rich compartments were finally defined as A, and the gene-poor compartments as B. For compartment display, we kept the absolute value of PC1, positive value represents compartment A, whereas negative value represents compartment B. Finally, considering a TAD is always contained in one compartment, we smoothed the compartmentation in TAD level. We assigned TADs to either the A- or the B- compartments, by calculating the average dominant eigenvector of each TAD. The 100 kb resolution contact matrices were transformed into an observed over expected (O/E) matrices with hicTransform [71] (part of hicexplorer, version 3.5.3, parameter: "--method obs_exp_lieberman") before calculating average contacts. Per-bin compartment strength was calculated by average A-A or B-B contact / average A-B contact of each bin. Saddleplots were drawn by cooltools compute-saddle (version 0.3.2, https://github.com/open2c/cooltools, parameter: "-n 50 --qrange 0.005 0.995") using 500 kb ICE-normalized O/E matrices. Homogeneity of A-A or B-B interactions was identified by the reciprocal of CV-square of average interactions between all A-type or B-type loci (CV, coefficient of variation). Compartment regions were determined by combining adjacent bins with same type.

### TAD analysis

TAD boundaries were identified by TopDom [74] (version 0.0.2, http://zhoulab.usc.edu/TopDom). To gain a better accuracy, we set the windows size gradient from 2 to 7 and found that a widow size of 5 have higher Pearson's correlation coefficient among all stages. Thus, we used TopDom with a window size of 5 from the 40 kb ICE-normalized matrices in this study. After getting the raw boundaries, we detected differential domain boundaries using diffHic [75] (version 1.22.0, https://anaconda.org/bioconda/bioconductor-diffhic). In details, we first converted the valid HiC interaction reads to HD5 file by each normalized replicate, and then calculated the Direction Index using domainDirections function setting width as 40 k and span as 7. Finally, we performed the significant difference test with replicates by glmQLFTest function, which integrated edgeR for dispersion estimation and GLM fitting. Only the raw TopDom TAD boundaries with FDR < 0.05 were treated as reorganization, and their corresponding TAD were classified as shift, fusion, and separation by homemade Perl script. The alluvial plots were drawn by R package ggalluvial (version 0.12.2, https://github.com/corybrunson/ggalluvial). Aggregate TAD analysis (ATA) was performed with 40 kb ICE-normalized matrices using GENOVA (version 0.9, https://github.com/robinweide/GENOVA). To analyze the reproducibility between the two replicates at each time point, we calculated genome-wide insulation score using GENOVA (version 0.9, https://github.com/robinweide/GENOVA) and calculated pairwise Pearson's correlation coefficient.

### Analysis of significant interactions

Hi-C contacts in untreated cells were aggregated between TADs to generate contact matrices of TAD pairs. Next, we calculated $p$ value of each interaction pair by a non-ventral hypergeometric distribution test and corrected for multiple testing by selecting interactions with FDR < 0.01 using the Benjamini–Hochberg method. Further, the interactions with O/E fold change lower than 10-fold were filtered out. Details on this statistical model have been published previously [76]. From these interactions, we identified "condensate-enriched interactions" wherein 40 kb bins at both ends overlapped with top 10% ChIP-seq signals of key components of specific condensates. As a control, we also identified the interactions between bins overlapping with bottom 10% signals as "condensate-absent interactions," as well as the interactions between randomly selected 10% bins as "random interactions." APA and quantification of interactions around the center of long-range condensate-enriched interactions was performed with 40 kb ICE-normalized matrices using GENOVA (version 0.9, https://github.com/robinweide/GENOVA) with default parameters in Fig. 7B. APA and quantification on structural factor-mediated interactions or E-P interactions were performed with 10 kb ICE-normalized matrices.

### Identification of chromatin loops

Chromatin loops of Hi-C were determined by cLoops [77] (v0.93, https://github.com/YaqiangCao/cLoops, parameters: "-eps 40000 -minPts 5 –hic").

### Visualization

Contact matrices were visualized using Juicebox [68] (version 1.11.08, https://www.aidenlab.org/juicebox). Tracks of PC1, compartment strength, and average contacts were plotted using hicPlotMatrix [71] (part of hicexplorer, version 3.5.3). Tracks of ChIP-seq data were plotted by Wash U Epigenome Browser (http://epigenomegateway.wustl.edu/browser/).

### ChIP-seq analysis

Fastq files were extracted from SRA using Sratoolkit (version 2.8.1, https://github.com/ncbi/sra-tools), trimmed adaptors by Cutadapt (version 2.4, http://code.google.com/p/cutadapt), and aligned to mm9 reference genome using Bowtie2 (version 2.2.5, http://bowtie-bio.sourceforge.net/bowtie2/) with default parameters. Reads with a mapq score less than 30, and PCR duplications were filtered out by using Samtools [78] (version 1.9, http://samtools.sourceforge.net). Reads aligned to the regions in ENCODE blacklist (http://mitra.stanford.edu/kundaje/akundaje/release/blacklists/) were discarded through bedtools [79] (version 2.29.1, https://bedtools.readthedocs.io/en/latest/). Fold enrichment analysis was performed same as in a previous study. In detail, we first binned the signals into 40 kb or 100 kb bins (taking the sum in each bin, 40 kb for TAD-related analysis, 100 kb for compartment-related analysis). Then, we calculated the median value of the signal track in bins within the cluster of interest and divided that by the median value of the signal track across all bins. To determine how the signal tracks correlated with the clusters, we calculated the Spearman correlation coefficient between the binned signal track and a pseudo cluster track, where the pseudo cluster track had 1 s at each bin that belonged to the cluster of interest and – 1 s at all other loci. We

also performed the Wilcoxon rank-sum test, comparing the values of the mark in loci within the cluster of interest to the values of the mark in all loci of the remaining clusters. Peaks were called with macs2 [80] (version 2.1.2, parameters: "-g mm -q 0.05 -m 5 50") using input as control. Peak density around boundaries was calculated using homemade Perl script. ATAC-seq data was processed same as ChIP-seq data. Enrichment analysis of Cohesin on anchors of interactions in Fig. S8 was performed by deepTools [81] (version 3.1.1, computeMatrix, https://deeptools.readthedocs.io/en/develop/).

### Gene ontology analysis

Gene symbols were first converted to EntrzID with R package BiomaRt [82] (version 2.42.0). "ensembl" was used as biomart database, and "mmusculus_gene_ensembl" was used as dataset. EntrzID of interest genes was uploaded to DAVID 6.8 [83] (https://david.ncifcrf.gov). Clustering of gene oncology enrichment was performed and visualized by pheatmap using Ward's method (version 1.0.12, https://cran.r-project.org/web/packages/pheatmap).

### Data availability

Hi-C data generated in this study are available at NCBI under accession number PRJNA665281 [84]. Raw reads of ChIP-seq data were downloaded from SRA SRP137821 (MED1, BRD4) [1], SRP097719 (HP1) [85], SRP162925 (SC35, RNAPII) [51], SRP108392 (Ring1B) [86], SRP121483 (H3K9me3) [87], SRP006786 (H3K27ac, H3K4me1, H3K4me3) [88], SRP017752 (H3K27me3, H3K9me2) [89], SRP048947 (H3K36me3) [90], and SRP004033 (mC, hmC) [91] and re-processed as described in "Materials and methods." Raw Hi-C data before and after CTCF-degradation were downloaded from SRP106652 [40] and re-processed as described in "Materials and methods" but skipped the step of trimLinker and used the restriction site of HindIII in the step of HiC-Pro. Raw reads of ATAC-seq data were downloaded from SRP094576 [92] and re-processed as described in "Materials and methods." Hi-C and RNA-seq data for mESCs were downloaded from GSE96107 [15].

### Supplementary Information

---

**Additional file 1: Fig. S1**. Controls of 1,6-HD treatment and effects of 1,6-HD treatment on cell viability, nuclear volume, chromatin mobility and chromatin condensation. **Fig. S2** 1.5%, 2 min 1,6-HD treatment dissolved biomolecular condensates in NPCs and MEFs without affecting cell viability and nuclear volume. **Fig. S3** Hi-C reproducibility across replicates and with published data. **Fig. S4** Extended analysis of compartmental changes and features of strengthened or weakened compartment loci after short-term 1,6-HD treatment. **Fig. S5** Features of compartment loci in different PC1 range and homogeneities of B-B interactions. **Fig. S6** Analysis of compartment switch and features of switched compartment loci. **Fig. S7** Extended analysis and validation of TAD changes and features of reorganized TAD boundaries after short-term 1,6-HD treatment. **Fig. S8** Comparison between condensate-component-enriched interactions and E-P interactions and clues for redundancy between condensate components.

**Additional file 2: Table S1**. Primers for ChIP-qPCR assays in Fig. S1E and 3C-qPCR assays in Fig. 4F and Fig. S7G-H.

**Additional file 3:.** Review history.

---

### Acknowledgements

We thank Prof. Lingling Chen, Shanghai Institute of Biochemistry and Cell Biology, Chinese Academy of Science, for kindly help with mRuby3-NONO-KI HeLa cells. We also thank Prof. Baohui Chen, Zhejiang University School of Medicine, for kindly help with H2B-GFP stably expressed HeLa Cells. Kindly, we thank Prof. Mingliang Zhang, Shanghai Jiao Tong University School of Medicine, for help with NPC cell line.

**Review history**

The review history is available as Additional file 3.

**Peer review information**

**Authors' contributions**

X.L. and S.J. contributed equally to this work. X.L., S.J., J.Q., and L.M. conceived the experiments. X.L., S.J., J.Q., L.M., X.Z., and L.Z. performed experiments. X.L. performed data analysis. X.L. and S.J. prepared the manuscript. J.D. supervised the project. All author(s) read and approved the final manuscript.

**Author's information**

Twitter handles: @Xinyi_Liu_671 (Xinyi Liu).

**Funding**

This research was funded by grants from the National Key Research and Development Program of China Stem Cell and Translational Research (2017YFA0102800 and 2016YFA 0101700), the National Natural Science Foundation of China (Grant Nos. 31970811 and 31771639), the Frontier Research Program of Bioland Laboratory (Guangzhou Regenerative Medicine and Health Guangdong Laboratory) (2018GZR110105007), and the Guangdong Innovative and Entrepreneurial Research Team Program 2016ZT06S029 to J. D.

## Declarations

**Ethics approval and consent to participate**

Not applicable.

**Competing interests**

The authors declare that they have no competing interests.

**Author details**

[1]RNA Biomedical Institute, Sun Yat-Sen Memorial Hospital, Zhongshan School of Medicine, Sun Yat-Sen University, Guangzhou 510080, China. [2]Center for Stem Cell Biology and Tissue Engineering, Key Laboratory for Stem Cells and Tissue Engineering, Ministry of Education, Sun Yat-Sen University, Guangzhou 510080, China. [3]Department of Cell Biology, Zhongshan School of Medicine, Sun Yat-Sen University, Guangzhou 510080, China. [4]Department of Histology and Embryology, School of Basic Medical Sciences, Guangzhou Medical University, Guangzhou 511436, China. [5]West China Biomedical Big Data Center, West China Hospital, Sichuan University, Chengdu 610041, China.

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

## 
