## [**Additional file 3:.** Review history. · Genome Biology]

Review History

First round of review

Reviewer 1

Are you able to assess all statistics in the manuscript, including the appropriateness of statistical tests used? No, I do not feel adequately qualified to assess the statistics.

Comments to author:

Xinyi Liu et al 2021, Genome Biology

In this study, Liu et al analyzed the effects of different concentrations and treatment durations of 1,6-HD, and found that 1.5%, 2min is a proper condition to dissolve condensates with minimal side effects in mESCs. Furthermore, they constructed maps of 3D chromatin organization following short- or long-term exposure to 1.5% 1,6-HD. Their results indicated that short-term treatment with 1.5% 1,6-HD resulted in reduced long-range interactions, strengthened compartmentalization, homogenized A-A interactions, B-to-A compartment switch and TAD reorganization, whereas longer exposure had the opposite effects.

As the authors mentioned that 1,6-HD is the only available tool to globally disrupt condensates, this study addresses an important technology that might be of broad interest to the field, phase transitions. However, I have the following questions:

1. There are so abundant biomolecular condensates within the cell, including the cytoplasm and the nucleus. Why is it crucial to address the global role played by condensates in 3D chromatin organization? Wouldn't it be more important to analyze a particular nuclear condensates?

2. The authors aimed to standardize the parameters of 1,6-HD treatment for all cell types. However, they only reported the results of one single cell type.

3. In the section of Discussion Line 10-19, they mentioned "Since the basic biochemical mechanism of protein condensation is common to all cell types, the standardized 1,6-HD condition can help analyze dissolution of condensates in other cell types as well, and in fact did show encouraging results in mouse embryonic fibroblasts (MEFs) and neural progenitor cells (NPCs) (data not shown)"

It might be difficult to find a standardized 1,6-HD condition for all cell types. The authors should show their results in MEFs and NPCs to support their conclusion.

4. Line 26-32 in Discussion, the authors summarized as follows :

"In this study we provided a time-resolved map of 3D genome during a 1,6-HD-induced phase transition process of nuclear condensates, which illustrated the 3D genome disorders accompanied by aberrant phase transitions."

1,6-HD is a tool to globally disrupt condensates, so the effect is not limited to nuclear

condensates. Does it make sense to explore the change of 3D chromatin organization under the disruption of all condensates in the cell?

Reviewer 2

Are you able to assess all statistics in the manuscript, including the appropriateness of statistical tests used? Yes, and I have assessed the statistics in my report.

Comments to author:

Liu et al. tried to benchmark the 1,6-HD condition and investigate nuclear condensates' roles in 3D chromatin organization are interesting. The 1,6-HD was prevented from being widely used as the working conditions are not standardized currently, and its molecular characterizations are unclear. The authors made efforts to identify a proper 1,6-HD condition is not a significant breakthrough in the field, but this does help provide useful information for the researcher to use this chemical or the readers to evaluate the data generated after 1,6-HD treatment in the literature. On the other side, it is quite interesting to investigate nuclear condensates' roles in 3D chromatin organizations systematically. However, the current manuscript's statement is hampered by lacking sufficient controls, both positive and negative controls, and their quantification also appears to be subtle, so this needs significant attention if there is a re-submission. Besides, the comprehension of the 3D chromatin organization analyses part is somewhat unclear by the paucity of enough information of the analyses and the data representation and interpretation.

Major:

1. Benchmark the conditions of 1,6-HD.

A major potential issue in the manuscript lacks sufficient controls to benchmark the 1,6-HD conditions. The 1,6-HD is used to disrupt hydrophobic interactions, and concerns were recently raised for this chemical in chromatin study (PMID: 33536240). The authors tried to standardize the 1,6-HD usage would be helpful for the community. Unfortunately, many controls and comparisons are missing in the current manuscript. The previous studies showed that 1,6-HD treatment decreases the mediator and BRD4 chromatin binding (PMID: 29930091; 33431820), how these bindings are affected in the proper 1,6-HD conditions. As this study investigated the roles in 3D chromatin organization, and the examination of CTCF and Cohesin binding is also required. As the formaldehyde crosslinking may create artificial effects of protein mislocalization or aggregation, it is necessary to perform the time-series analyses of 1,6-HD treatment with the endogenous tagged nuclear condensate proteins.

The authors claimed that they got encouraging results in MEFs and NPCs with the 1.5%, 2min 1,6-HD treatment, but it is not shown. These data need to show. To standardize the 1,6-HD treatment, the 2min treatment is not very practical for bulk analyses, as it takes time for the chemical to permeate into the cells and nucleus, and it is also quite variable among the cells if the time is too short. To standardize the 1,6-HD condition, the author may consider lowering the concentration and increasing the treatment time (maybe 10min to 30min). On the other side, the information of the 1,6-HD used in this study cannot be found in the method section, and the comparison 1,6-HD produced from different resources would also help standardize the 1,6-HD treatment.

2. The structural analyses need positive, negative controls and validations.

The authors performed the compartment strength (Figure 3A-B), compartment interactions (Figure 4D-E), compartment switch (Figure 5A-B), TAD length analyses (Figure 6A-D), and condensates-enriched interactions (Figure 7) analyses. However, most of the changes are marginal and hard to be convincing. For example, the CTCF depletion does not affect the chromatin compartmentalization. It is necessary to perform the parallel compartment analyses with the CTCF degron Hi-C data (PMID: 28525758). The WAPL knockout increases the loop or TAD length (PMID: 28475897). The SPRITE and TAS-Seq have identified the nuclear speckles and nucleolus-associated chromatin interactions (PMID: 29887377). The structure-related analyses need to use these published datasets to further evaluate the current manuscript's changes. On the other side, it would be convening to perform the analyses with each replicate separately instead of the merged data, and the validations of the Compartment or TAD changes are also required with independent experimental analyses rather than only Hi-C datasets currently.

Minor:

Figure 1B. For the time-series imaging analyses, negative controls are required. Such as the proteins that do not form nuclear condensates, CTCF and Cohesin would be interesting candidates, as they are directly involved in 3D chromatin organization. On the other side, it is strange that the most prominent nuclear condensate nucleolus was also not examined.

Figure 2C. The interactions in the black box appear to increase first, then decrease as claimed by the authors. However, why the interaction frequency at the other loop anchors seem to decrease first? It is also hard to evaluate the chromatin interaction changes with the current presentation, the quantification of the changed percentage with different replicates, and quantification of interactions of an adjacent loop anchor as a control.

Figure 6G-H. What did the black lines indicate in the figure? Are they TADs? It is very hard to appreciate that they are TADs. How did the chromatin domains look like the same regions with the high-resolution Hi-C data published elsewhere?

Figure 7B. The chromatin interactions are usually less frequent in the regions besides CTCF and Cohesin binding sites. It is pretty strange that the APA signals are quite high in Figure 7B. The detailed quantification methods should be provided, and APA signals at the CTCF and Cohesin binding sites would also be helpful to represent in parallel.

P7 line 1. The 5% and 10% 1,6-HD data were not shown in Figure 1A.

P7 line 26. A reference is needed for the statement, "A previous study showed that the reappearance of protein aggregates is likely due to cell shrinkage and increased macromolecular crowding."

Discussion section, P15, line 41. The authors mentioned that the two preprints also used the 1,6-HD to investigate nuclear condensates' roles in 3D chromatin organization. The author needs to

discuss why high concentration and low concentration treatment are opposite? What are the biological implications for this? What are the dangers of using a high concentration of 1,6-HD? Is it also consistent with results presented in two preprints?

Method section, P24, line 20, Do "RYBP" and "CTCF" were shown in the current work?

The authors need to provide enough details about how they perform the puncta analyses.

Reviewer 3

Are you able to assess all statistics in the manuscript, including the appropriateness of statistical tests used? No, I do not feel adequately qualified to assess the statistics.

Comments to author:

In this paper, the authors use 1,6-HD to explore the possible role of phase condensate formation on chromatin structure. First, they treat ES cells with different amounts of 1,6-HD for different times, and choose a concentration (1.5%) that is less destructive to the cell. They then use HiC to show that 1,6-HD treatment causes changes in TAD structures, including decreasing long range interactions and increasing short range interactions.

Major concerns

1. The authors talk about phase condensates as though it is completely accepted in the field that these exist in the cell and that they mediate important chromatin interactions. Multiple papers have questioned the existence and/or importance of phase condensates, especially for transcription, chromatin structure or super-enhancer function (see McSwiggen et al G&D (2019), PMID: 31594803 for an excellent review on this topic). The problem with this, is that even if 1,6-HD dissolves phase condensates in vitro, it is hard to prove what it is actually doing to a living cell, especially if BRD4/Mediator phase condensates turn out to be an artifact. So really, this paper is about what 1,6-HD does to chromatin, but if 1,6-HD is a non-specific tool, then it is difficult to draw any biologically meaningful conclusions from the data. What is needed is an independent technique for dissolving phase condensates that is not dependent on 1,6-HD treatment, in order to validate these findings.

2. Further to this, the authors have failed to reference an important study in this area (Itoh et al, Life Science Alliance (2021), PMID: 33536240). Itoh et al used Halo-tagged H2B and live imaging to measure chromatin changes and found that chromatin essentially becomes "fixed" at high concentrations of 1,6-HD, similar to the results of long exposure reported here. Itoh et al also find that even at lower doses (2.5% for 5 minutes), nucleosome motion is severely reduced, which suggests that 1,6-HD has a non-specific impact on chromatin mobility. I understand that the authors in this study use an even lower dose of 1,6-HD (1.5%), but it seems quite possible that even at 1.5%, 1,6-HD impacts chromatin mobility in a non-specific manner that has nothing to do with dissolving phase condensates. This non-specific effect could explain the results observed here, but it doesn't accomplish what the authors hope to do, which is to link phase condensate formation to higher order chromatin structure.

3. Further, if we accept for a moment that proteins such as BRD4 and Mediator actually do form phase condensates in the cell, two recent studies cast doubt on the idea that these proteins have much of an impact on chromatin structure. El Khattabi et al used a degron for Mediator along with HiC to show that Mediator is mostly dispensible for chromatin interactions, although they did see a very slight increase in interactions (El Khattabi et al, Cell (2019) PMID: 31402173). More recently, using a degron for BRD4 (as well as BET inhibitors) and a high-resolution 3C method called Capture C, Crump et al showed that enhancer promoter interactions remain intact in the absence of both BRD4 and Mediator, although very slight increases in interactions were also observed (Crump et al, Nat Comm (2021) PMID: 33431820). However, these increased interactions varied between treatments and did not correlate well with gene regulation, so they are likely biologically meaningless. If BRD4 and Mediator form phase condensates in the cell, but degrading these proteins doesn't have an impact on chromatin structure, this casts doubt on whether 1,6-HD mediated chromatin changes are actually due to disrupting phase condensate formation. Instead, it suggests that the changes observed here that are induced by 1,6-HD could just be an off-target effect.

We appreciate the constructive and in-depth comments and suggestions of the three reviewers, which helped us to improve the study and the manuscript substantially. We have followed their suggestions and performed numerous additional experiments and analyses (Fig.1B, Fig.2C, Fig. 7D-E, Fig. S1B-C, Fig. S1E-J, Fig. S2, Fig. S4A-H, Fig. S5A-B, Fig. S6B-C, Fig. S7A-H and Fig.S8 in the Revised MS). In this letter, we provided detailed point-by-point responses to the reviewers' comments together with our additional information and new experimental data.

Reviewer #1:

In this study, Liu et al analyzed the effects of different concentrations and treatment durations of 1,6-HD, and found that 1.5%, 2min is a proper condition to dissolve condensates with minimal side effects in mESCs. Furthermore, they constructed maps of 3D chromatin organization following short- or long-term exposure to 1.5% 1,6-HD. Their results indicated that short-term treatment with 1.5% 1,6-HD resulted in reduced long-range interactions, strengthened compartmentalization, homogenized A-A interactions, B-to-A compartment switch and TAD reorganization, whereas longer exposure had the opposite effects.

As the authors mentioned that 1,6-HD is the only available tool to globally disrupt condensates, this study addresses an important technology that might be of broad interest to the field, phase transitions. However, I have the following questions:

Response: Thanks for your positive comments and constructive suggestions to advance our work. We have carefully responded your comments and questions as follows.

1. There are so abundant biomolecular condensates within the cell, including the cytoplasm and the nucleus. Why is it crucial to address the global role played by condensates in 3D chromatin organization? Wouldn't it be more important to analyze a particular nuclear condensate?

Response: Thank you for asking the important questions. The necessity and importance of studying the global role played by condensates in 3D chromatin organization mainly lies in three aspects:

- a) First of all, although nuclear condensates have recently been proposed to have a prominent role in chromatin organization¹, the detailed relationship remains largely unclear. At this aspect, a global view of 3D chromatin reorganization after dissolving condensates should help to figure out possible models of the detailed molecular mechanisms and provide clues for further investigation of particular nuclear condensate.
- b) Secondly, although a few studies have suggested that particular nuclear condensates play important role in 3D chromatin organization (including speckles² and OCT4³), these studies failed to provide a global overview of the relationship between nuclear condensates and 3D chromatin organization. A profile of the 3D chromatin organization after dissolving condensates are still lack in the field.
- c) Furthermore, the dynamic profile of 3D chromatin organization during 1,6-HD treatment is a resource, which can be used for preliminarily exploring the relationship between particular nuclear condensates and 3D chromatin organization by integrated analysis with ChIP-seq of components of particular condensate.

2. The authors aimed to standardize the parameters of 1,6-HD treatment for all cell types. However, they only reported the results of one single cell type.

Response: Thanks for the important comment. We tested the parameters of 1,6-HD treatment in NPCs and MEFs and included these data in the revised manuscript (**Revised MS, Fig. S2**), the details are shown in the answer of the next point.

3. In the section of Discussion Line 10-19, they mentioned "Since the basic biochemical mechanism of protein condensation is common to all cell types, the standardized 1,6-HD condition can help analyze dissolution of condensates in other cell types as well, and in fact did show encouraging results in mouse embryonic fibroblasts (MEFs) and neural progenitor cells (NPCs) (data not shown)" It might be difficult to find a standardized 1,6-HD condition for all cell types. The authors should show their results in MEFs and NPCs to support their conclusion.

Response: Thank you for the comment and constructive suggestion. Although it might be difficult to find a standardized 1,6-HD condition for all cell types, we have proved that the proper

condition of 1,6-HD we found in mESCs (1.5%, 2min) also works in other cell types, including NPCs, MEFs and HeLa. In detail, this condition also dissolved biomolecular condensates without impairing cell viability and decreasing nuclear volume in NPCs and MEFs (**Response R1 Fig. 1**). Besides, we also observed a dissolution of paraspeckles upon 1.5%, 2min 1,6-HD by live cell imaging (**Revised MS, Fig. S1C**). These results further support that 1.5%, 2min is a proper condition of 1,6-HD treatment which may be suitable for various cell types. We have added these results in new **Fig. S2** (descriptions in **Revised MS, Page 8**).

Response R1 Fig. 1: 1.5%, 2min 1,6-HD treatment dissolved biomolecular condensates in NPCs and MEFs without affecting cell viability and nuclear volume. (A-B) Left: SIM images of IF for the proteins indicated by green words in NPCs (A) or MEFs (B). IF for indicated protein is colored green, and signal from DAPI is colored dark blue. Right: Counts of IF puncta within nucleus at each time points. All p values were determined using the Student's t test. (C-D) Apoptosis of NPCs (C) or MEFs (D) upon 1,6-HD treatment. (E-F) Left: SIM images of DAPI-

stained NPCs (E) or MEFs (F). Right: quantification of nuclear volume. All p values were determined using the Wilcoxon rank-sum test.

4. Line 26-32 in Discussion, the authors summarized as follows: "In this study we provided a time-resolved map of 3D genome during a 1,6-HD-induced phase transition process of nuclear condensates, which illustrated the 3D genome disorders accompanied by aberrant phase transitions." 1,6-HD is a tool to globally disrupt condensates, so the effect is not limited to nuclear condensates. Does it make sense to explore the change of 3D chromatin organization under the disruption of all condensates in the cell?

Response: Thank you for the question. Indeed, it is hard to rule out whether the disruption of condensates in the cytoplasm or membrane after 1,6-HD treatment has an impact on 3D chromatin organization. Thus, we replaced the word "biomolecular" with "nuclear" and improved the sentence in Discussion as follows:

"In this study we provided a time-resolved map of 3D genome during a 1,6-HD-induced phase transition process of condensates, which illustrated the 3D genome disorders accompanied by aberrant phase transitions." (Revised MS, Page 19)

Rigorously, we replaced the word "nuclear condensates" with "biomolecular condensates" in the revised manuscript.

Nevertheless, it still makes sense to study the change of 3D chromatin organization upon 1,6-HD treatment. We did find chromatin organization changes are related to neighboring nuclear condensates through integrated analysis of ChIP-seq data of components of condensates. Particularly, the interactions between the condensate-component-enriched regions were greatly affected by 1,6-HD treatment (Revised MS, Fig. 7).

Reviewer #2:

Liu et al. tried to benchmark the 1,6-HD condition and investigate nuclear condensates' roles in 3D chromatin organization are interesting. The 1,6-HD was prevented from being widely used as the working conditions are not standardized currently, and its molecular characterizations are unclear. The authors made efforts to identify a proper 1,6-HD condition is not a significant breakthrough in the field, but this does help provide useful information for the researcher to use this chemical or the readers to evaluate the data generated after 1,6-HD treatment in the literature. On the other side, it is quite interesting to investigate nuclear condensates' roles in 3D chromatin organizations systematically.

Response: Thank you for the positive comments and the constructive suggestions to improve our manuscript. We have carefully responded your comments and concerns as follows.

However, the current manuscript's statement is hampered by lacking sufficient controls, both positive and negative controls, and their quantification also appears to be subtle, so this needs significant attention if there is a re-submission. Besides, the comprehension of the 3D chromatin organization analyses part is somewhat unclear by the paucity of enough information of the analyses and the data representation and interpretation.

Response: Thanks for the constructive comments. Following your suggestions, we performed additional experiments and analysis and largely revised our manuscript (**Revised MS, Fig.1B, Fig.2C, Fig. 7D-E, Fig. S1B-C, Fig. S1E-J, Fig. S2, Fig. S4A-H, Fig. S5A-B, Fig. S6B-C and Fig. S7A-H**). The details will be shown point-by-point as follows.

Major:

1. Benchmark the conditions of 1,6-HD.

A major potential issue in the manuscript lacks sufficient controls to benchmark the 1,6-HD conditions. The 1,6-HD is used to disrupt hydrophobic interactions, and concerns were recently raised for this chemical in chromatin study (PMID: 33536240). The authors tried to standardize the 1,6-HD usage would be helpful for the community. Unfortunately, many controls and comparisons are missing in the current manuscript.

Response: Thanks for the constructive comments. Although there is no appropriate control at

present that can completely rule out the side effects of 1,6-HD, we still followed your suggestions and exhausted all available methods to rule out the alternative possibilities:

a) First, in order to rule out that 1,6-HD indiscriminately affects the distribution patterns of all proteins, proteins that do not form condensates, including CTCF, SMC1A and Lamin B1, were tested as control. As expected, we found that the distribution patterns of neither SMC1A nor Lamin B1 was affected by 1,6-HD treatment (**Response R2 Fig. 1**). However, we observed that CTCF formed puncta in mESCs, which has been reported in previous studies⁴⁻⁶. We found that the puncta of CTCF were similarly disrupted by time-series 1,6-HD treatment as other nuclear condensates (**Response R2 Fig. 1**). Together, these results demonstrated that 1,6-HD has a time-dependent effect on condensates without affecting the distribution pattern of proteins that do not form condensates. We have added these results in new **Fig. S1B** (descriptions in **Revised MS, Page 5**).

Response R2 Fig. 1: 1.5% 1,6-HD treatment has a time-dependent effect on CTCF without affecting the distribution pattern of SMC1A and Lamin B1. Left: SIM images of IF for the proteins indicated by green words. IF for indicated protein is colored green, and signal from DAPI is colored dark blue. Right: Counts of IF puncta within nucleus at each time points versus 0min. Error bars represent standard deviation (SD) from 30 cells each. All p values were determined using the Student's t test.

b) Second, we have tried to include 2,5-HD, a chemical structurally similar to 1,6-HD but with less condensate-dissolving activity as a negative control of 1,6-HD^{7,8}. However, although

cell viability and many types of nuclear condensates remain unaffected upon 2,5-HD treatment, condensates of BRD4 and MED1 were largely dissolved (**Response R2 Fig. 2A-B**). These results suggest that 2,5-HD is not suitable for being used as negative controls of 1,6-HD in global level studies. We have added these results in new **Fig. S11-J** (descriptions in **Revised MS, Page 7**).

Response R2 Fig. 2: 1.5% 2,5-HD treatment does not affect cell viability but largely dissolve BRD4 and MED1 condensates. (A) Apoptosis analysis of mESCs upon 2,5-HD treatment. (B) Left: SIM images of IF for the proteins indicated by green words. IF for indicated protein is colored green, and signal from DAPI is colored dark blue. Right: Counts of IF puncta within nucleus at each time. All p values were determined using the Student's t test.

c) Third, in order to eliminate the influence of formaldehyde cross-linking, we performed live-cell fluorescence microscopy of endogenous tagged paraspeckles (indicated by NONO⁹). As a result, we observed a similar time-dependent effect of 1,6-HD with the results in fixed cells, where short-term exposure to 1.5% 1,6-HD dissolved condensates and long-term exposure caused re-aggregation (**Response R2 Fig. 3**). These results further confirmed the time-dependent effect of 1,6-HD. We have added these results in new **Fig. S1C** (descriptions in **Revised MS, Page 5**).

Response R2 Fig. 3: 1.5% 1,6-HD treatment has a time-dependent effect on paraspeckles in live cell. Live-cell fluorescence microscopy of endogenous tagged paraspeckles (indicated by NONO). The white arrows indicate observed condensates.

d) Finally and importantly, we have carefully read the article you mentioned (PMID: 33536240, which was not online when we submitted the manuscript). This study has reported that 5% or higher concentration 1,6-HD treatment for 5min had a side effect of suppressing chromatin motion and hyper-condensing chromatin in live cells¹⁰. We therefore examined the effects of the proper condition we optimized in our work (1.5%, 2min) on chromatin behavior. We performed super-resolution live-cell imaging on stably expressed histone H2B-GFP in HeLa cells and recorded images at 50ms/frame (~300 frame, 15s total) (**Response R2 Fig. 4A**). Notably, trajectory analysis revealed that although 10%, 5min 1,6-HD drastically suppressed chromatin motion as reported, 1.5%, 2min 1,6-HD did not affect chromatin motion (**Response R2 Fig. 4B**). Besides, we analyzed the distribution of H2B used the L-function, which quantitates the size of H2B clusters and the degree of accumulation. As a result, 10%, 5min 1,6-HD caused chromatin hyper-condensation as reported, while 1.5%, 2min 1,6-HD exhibited a similar H2B clustering to untreated cells (**Response R2 Fig. 4C**). These results demonstrated that 1.5%, 2min 1,6-HD does not have the side effect of suppressing chromatin motion and hyper-condensing chromatin, which further supported that it is a proper using condition. We have added these results in new **Fig. S1F-H** (descriptions in **Revised MS, Page 7**).

Response R2 Fig. 4: 1.5%, 2min 1,6-HD treatment does not have the side effect of suppressing chromatin motion and hyper-condensing chromatin. (A) Super-resolution live-cell imaging of stably-expressed H2B-GFP. **(B)** Mean square displacement plots indicating chromatin motion (\pm SD among cells). All p values were determined using the Kolmogorov-Smirnov test. **(C)** L-function plot indicating chromatin condensation.

Taken together, we have additionally included more controls and comparisons in our revised manuscript, which reinforced our claims that 1.5%, 2min is a proper condition of 1,6-HD treatment.

The previous studies showed that 1,6-HD treatment decreases the mediator and BRD4 chromatin binding (PMID: 29930091; 33431820), how these bindings are affected in the proper 1,6-HD conditions. As this study investigated the roles in 3D chromatin organization, and the examination of CTCF and Cohesin binding is also required.

Response: Thank you for this constructive suggestion. We have performed the ChIP-qPCR assay in mESCs on the sites shown as examples in the article you mentioned^{11,12} (**Response R2 Fig. 5**). The results showed that:

- a) First, the binding of MED1 decreased dramatically and no effect on BRD4 binding at the sites were observed after the 1,6-HD treatment, which were comparable to the ChIP-seq data from the previous studies at those loci¹¹.
- b) Second, CTCF binding was up-regulated at these loci. Previous studies showed that CTCF exhibit as cluster⁴⁻⁶. We also observed the puncta formation and the sensitive characteristics to 1,6-HD of CTCF. These results implied that the dissolution of CTCF clusters induced by 1,6-HD may have a complicated impact on CTCF binding.
- c) Third, we did not observe the obvious aggregation of Cohesin and Coheisn binding was

mainly unchanged.

We have added these results in new Fig. S1E (descriptions in Revised MS, Page 6).

Response R2 Fig. 5: Effects of 1,6-HD treatment on chromatin binding of BRD4, MED1, CTCF and SMC1A. Top: tracks of published ChIP-seq data. The y axis shows reads per kilobase per million reads (RPKM). Bottom: Relative enrichment of indicated factors determined by ChIP-qPCR assay at indicated sites. All p-values were determined by the Student's t test. Three biological replicates were assayed in ChIP-qPCR assays.

As the formaldehyde crosslinking may create artificial effects of protein mislocalization or aggregation, it is necessary to perform the time-series analyses of 1,6-HD treatment with the endogenous tagged nuclear condensate proteins.

Response: Thank you for this constructive suggestion. As detailedly described in Response Letter Page 8 and showed in Response R2 Fig. 3, we performed live cell imaging of the endogenous tagged paraspeckles and observed a similar time-dependent effect of 1,6-HD with the results in fixed cells, which further confirmed the time-dependent effect of 1,6-HD (Response R2 Fig. 3). We have added these results in new Fig. S1C (descriptions in Revised MS, Page 5).

The authors claimed that they got encouraging results in MEFs and NPCs with the 1.5%, 2min 1,6-HD treatment, but it is not shown. These data need to show.

Response: Thanks for your suggestion. We have included the data of MEFs and NPCs in the revised manuscript. Notably, 1.5%, 2min 1,6-HD treatment also dissolved nuclear condensates without impairing cell viability and decreasing nuclear volume in those two types of cells (Response R2 Fig. 6). These results further support that 1.5%, 2min is a proper condition of 1,6-HD treatment which may be suitable for various cell types. We have added these results in new Fig. S2 (descriptions in Revised MS, Page 8).

Response R2 Fig. 6: 1.5%, 2min 1,6-HD treatment dissolved biomolecular condensates in NPCs and MEFs without affecting cell viability and nuclear volume. (A-B) Left: SIM images of IF for the proteins indicated by green words in NPCs (A) or MEFs (B). IF for indicated protein is colored green, and signal from DAPI is colored dark blue. Right: Counts of IF puncta within nucleus at each time points. All p values were determined using the Student's t test. (C-D) Apoptosis of NPCs (C) or MEFs (D) upon 1,6-HD treatment. (E-F) Left: SIM images of DAPI-stained NPCs (E) or MEFs (F). Right: quantification of nuclear volume. All p values were determined using the Wilcoxon rank-sum test.

To standardize the 1,6-HD treatment, the 2min treatment is not very practical for bulk analyses, as it takes time for the chemical to permeate into the cells and nucleus, and it is also quite variable among the cells if the time is too short. To standardize the 1,6-HD condition, the author may consider lowering the concentration and increasing the treatment time (maybe 10min to 30min).

Response: Thanks for raising the concern and the valuable suggestion. In this study, in order to avoid variability of treatment among the cells, the ESC medium contained 1.5% 1,6-HD was fully mixed and warmed in 37°C incubator before treatment. Soon after discarding the medium in the dish, the homogeneous and warmed 1.5% 1,6-HD mixture was added and the dish was put back into the incubator immediately. We have added these details in the Materials and Methods part (**Revised MS, Page 26**).

Nevertheless, it is a great idea to lower the concentration and increasing the treatment time. We have tested the effects of lower concentrations (0.5% and 1%) for 30min, respectively. Unfortunately, although puncta number of BRD4 significantly decreased after treatment, the condensates of HP1a remained unaffected (**Response R2 Fig. 7**). Therefore, more optimized conditions with lower concentration and increased treatment time need to be further explored.

Response R2 Fig. 7: Lowering the concentration and increasing the duration of 1,6-HD treatment failed to dissolve condensates of HP1 α . Left: SIM images of IF for the proteins indicated by green words. IF for indicated protein is colored green, and signal from DAPI is colored dark blue. Right: Counts of IF puncta within nucleus at each time points. All p values were determined using the Student's t test.

On the other side, the information of the 1,6-HD used in this study cannot be found in the method section, and the comparison 1,6-HD produced from different resources would also help standardize the 1,6-HD treatment.

Response: Sorry for the omission. The information of 1,6-HD used in this study is Sigma, 240117. We have added this information in the Materials and Methods part (**Revised MS, Page 26**).

2.The structural analyses need positive, negative controls and validations.

The authors performed the compartment strength (Figure 3A-B), compartment interactions (Figure 4D-E), compartment switch (Figure 5A-B), TAD length analyses (Figure 6A-D), and condensates-enriched interactions (Figure 7) analyses. However, most of the changes are marginal and hard to be convincing. For example, the CTCF depletion does not affect the chromatin compartmentalization. It is necessary to perform the parallel compartment analyses with the CTCF degron Hi-C data (PMID: 28525758). The WAPL knockout increases the loop

or TAD length (PMID: 28475897). The SPRITE and TAS-Seq have identified the nuclear speckles and nucleolus-associated chromatin interactions (PMID: 29887377). The structure-related analyses need to use these published datasets to further evaluate the current manuscript's changes.

Response: Thanks for the constructive suggestions. We performed additional analysis to further evaluate our findings:

- a) In the aspect of the compartmental analysis, we compared the compartmental changes between 1.5%, 2min 1,6-HD treatment and CTCF depletion¹³, which does not affect compartmentalization. Parallel compartment analyses of our manuscript using the published data were performed and as expected, compartment strength and intra-compartmental interactions were more affected by 1,6-HD than CTCF degradation (Response R2 Fig. 8A-B). Besides, we also performed parallel compartment analysis as the CTCF-degron study¹³. Consistently, while contact maps and compartment signal (PC1) are maintained after CTCF depletion, 1.5%, 2min 1,6-HD exhibited a greater impact (Response R2 Fig. 8C-E). Besides, analysis of compartment strength as described in the the CTCF-degron study also revealed a reproducible increase in the strength of compartmentalization upon 1.5%, 2min 1,6-HD treatment (Response R2 Fig. 8F). These results further support that 1.5%, 2min 1,6-HD treatment did cause compartmental changes. We have added these results in new Fig. S4C-H (descriptions in Revised MS, Page 9).
- b) In the aspect of TAD analysis, we performed parallel analysis as the WAPL knock-out study¹⁴ (Response R2 Fig. 8G). We respectively quantified the length of chromatin loops in compartment A or B upon 2min 1,6-HD treatment. As a result, we observed that length of loops enlarged in compartment A while shortened in compartment B, which is consist with the changes of TAD sizes (Response R2 Fig. 8H-I). These results further support the findings of changed TAD sizes upon 1.5%, 2min 1,6-HD treatment. We have added these results in new Fig. S7E-F (descriptions in Revised MS, Page 13).
- c) In the aspect of the nuclear speckles and nucleolus-associated chromatin interactions, we have reprocessed the SPRITE data to identify the reported inter-chromosomal nuclear speckle hub and nucleolar hub and performed APA analysis respectively on these two hubs

in our data (Response R2 Fig. 8J-K). However, we failed to detect those inter-chromosomal interactions. This is probably because of technical differences between SPRITE and Hi-C that SPRITE can detect many inter-chromosomal interactions that cannot be observed by Hi-C, as reported in the SPRITE study¹⁵.

Response R2 Fig. 8: Parallel analysis with published data or as published studies. (A) Compartment strength of all 100kb bins from A- or B- compartment types. **(B)** Average interactions between strong A-featured loci (high PC1 A, $PC1 > 0.8$) or weak A-featured loci versus untreated cells. **(C)** Contact maps at across Chr2. Bars below denotes compartment A (red) or B (blue) at 100kb resolution. **(D)** Distribution of PC1 at 100kb resolution across Chr2. **(E)** Pairwise correlations of eigenvectors (PC1) at 100kb resolution. **(F)** Quantification of

compartmentalization strength as Nora et al. 2017. P values were determined by paired t -test. (G-I) Density plot showing the length distribution of the chromatin loops. (J-K) Left: Circos diagram of the indicated hubs identified by SPRITE. Right: APA analysis at the indicated hubs in our data (0min).

On the other side, it would be convening to perform the analyses with each replicate separately instead of the merged data,

Response: Thank you very much for this constructive suggestion. We reanalyzed our data in different replicates including the contacts between compartments (Response R2 Fig. 9A), the compartment strength (Response R2 Fig. 9B), the interactions within A compartments (Response R2 Fig. 9C-D), the compartment sizes (Response R2 Fig. 9E-F) and TAD sizes (Response R2 Fig. 9G-H). As a result, the effects of 1,6-HD treatment on different hierarchies of 3D chromatin organization were also observed in biological replicate similar to the merged data, which further reinforced our findings. We have added these results in new Fig. S4A-B, Fig. S5A-B, Fig. S6B-C and Fig. S7C-D respectively (descriptions in Revised MS, Page 9, 11, 12 and 13 respectively).

Response R2 Fig. 9: Effects of 1,6-HD treatment were also observed in biological replicate. (A) Average contacts between 100kb bins from the same (A-A, B-B) and different (A-B) compartment types at each time point in different biological replicates. (B) Compartment strength of all 100kb bins from A-, B- and both compartment types at each time in different biological replicates. (C-D) Average interactions between strong A-featured loci (high PC1 A, $PC1 > 0.8$) or weak A-featured loci versus untreated cells in different biological replicates. (E-F) Average size of compartment A (E) or B (F) regions at each time point in different biological replicates. (G-H) Average size of TADs in compartment A (G) or compartment B (H) at each time point in different biological replicates.

and the validations of the Compartment or TAD changes are also required with independent experimental analyses rather than only Hi-C datasets currently.

Response: Thank you very much for the valuable suggestion. We confirmed our findings in compartment and TAD changes by 3C-qPCR assay. As a result, at compartmental level, the decreased interactions between genomic loci with stronger A-features and the increased interactions between genomic loci with weaker A-features were further confirmed (Response R2 Fig. 10A). At TAD level, interactions between fused TADs in A compartment significantly increased (Response R2 Fig. 10B) and interactions within separated TADs in B compartment significantly decreased (Response R2 Fig. 10C), consist with the observation in Hi-C data. These results further confirmed our findings of compartment and TAD changes. We have added these results in new Fig. 4 and Fig. S7 (descriptions in Revised MS, Page 10 and 13).

Response R2 Fig. 10: Validations of the Compartment or TAD changes by 3C-qPCR assay. (A) Top: Example (chr3: 80-140Mb) showing homogenized A-A interactions at 2min. Top left: PC1 indicates compartment assignments at 0min and 2min (A, red; B, blue). Center: heatmap showing difference of observed contacts between 2min and 0min (2min minus 0min). Bottom: fold change of strength of indicated interactions determined by 3C-qPCR assay. All p-values were determined by the Student's t test. Three biological replicates were assayed for 3C-qPCR experiment. (B-C) Top: PC1 indicates compartment assignment and contact maps at indicated time point. Bottom: fold change of strength of indicated interactions determined by 3C-qPCR assay. All p-values were determined by the Student's t test. Three biological replicates were assayed for 3C-qPCR experiment.

Minor:

Figure 1B. For the time-series imaging analyses, negative controls are required. Such as the proteins that do not form nuclear condensates, CTCF and Cohesin would be interesting candidates, as they are directly involved in 3D chromatin organization.

Response: Thank you for the valuable suggestions. As detailedly described in Response Letter Page 7 and showed in Response R2 Fig. 1, we performed time-series imaging analyses on proteins CTCF, SMC1A and Lamin B1 and found that 1,6-HD treatment did not affect the distribution patterns of SMC1A and Lamin B1, while had a time-dependent effect on CTCF clusters⁴⁻⁶ (Response R2 Fig. 1). We have added these results in new Fig. S1B (descriptions in Revised MS, Page 6).

On the other side, it is strange that the most prominent nuclear condensate nucleolus was also not examined.

Response: Thanks for the comment. We additionally performed the time-series imaging analyses of nucleoli (indicated by Nucleolin) and observed the same time-dependent effect as other nuclear condensates (Response R2 Fig. 11). We have added these results in new Fig. 1B (descriptions in Revised MS, Page 5).

Response R2 Fig. 11: 1.5%, 1,6-HD has a time-dependent effect on nucleoli. Left: Structured illumination microscopy (SIM) images of immunofluorescence (IF) for the proteins indicated by green words in mESCs. IF for indicated protein is colored green, and signal from DAPI is colored dark blue. Right: Counts of IF puncta within nucleus at each time points versus 0min. Error bars represent standard deviation (SD) from 30 cells each. All p values were determined using the Student's t test.

Figure 2C. The interactions in the black box appear to increase first, then decrease as claimed by the authors. However, why the interaction frequency at the other loop anchors seem to decrease first? It is also hard to evaluate the chromatin interaction changes with the current presentation, the quantification of the changed percentage with different replicates, and quantification of interactions of an adjacent loop anchor as a control.

Response: Thanks for the question and the valuable suggestion. Indeed, there are short-range interactions decreased first. However, we would like to restate that the conclusion that short-range interactions (around 100kb) increased upon short-term 1,6-HD exposure and decreased upon long-term exposure came from analysis of relative contact probabilities (RCP). The increase and decrease of RCP did be significant across all chromosomes (Fig. 2B). Besides, following your constructive suggestions, we quantified changed percentage of interactions across certain genomic with different biological replicates, compared to the average interaction strength between the same bin pair in the two replicates of 0min (Response R2 Fig. 12). There did be a portion of short-range interactions cross 100kb decreased first upon 1,6-HD treatment as you observed in the example differential heatmap, but the larger portion of interactions at this range increased first, which is consist with the results of relative contact frequency analysis. In addition, we also tried to use the interactions between adjacent bins as a control but found they were also affected by 1,6-HD treatment. We have added these results in new Fig. 2C (descriptions in Revised MS, Page 8).

Response R2 Fig. 12: 1.5%, 1,6-HD has a time-dependent effect on nucleoli. Left: Structured illumination microscopy (SIM) images of immunofluorescence (IF) for the proteins indicated by green words in mESCs. IF for indicated protein is colored green, and signal from DAPI is colored dark blue. Right: Counts of IF puncta within nucleus at each time points versus 0min. Error bars represent standard deviation (SD) from 30 cells each. All p values were determined using the Student's t test.

Figure 6G-H. What did the black lines indicate in the figure? Are they TADs? It is very hard to appreciate that they are TADs. How did the chromatin domains look like the same regions with the high-resolution Hi-C data published elsewhere?

Response: We identified TADs by TopDom, a tool known to be robust to resolution and sequence depth¹⁶. The identification can be verified by comparison to the contact maps from ultra-deep in situ Hi-C data¹⁷ (Response R2 Fig. 13A) and the high density of CTCF peaks on TAD boundaries (Response R2 Fig. 13B). We have added these results in new Fig. S7A-B (descriptions in Revised MS, Page 13).

Response R2 Fig. 13: Validation of TAD identification. (A) Contact maps showing TADs in high-resolution Hi-C data and 0min. (B) Density of CTCF ChIP-seq peaks in identified TAD boundaries compared with shuffled regions.

Figure 7B. The chromatin interactions are usually less frequent in the regions besides CTCF and Cohesin binding sites. It is pretty strange that the APA signals are quite high in Figure 7B. The detailed quantification methods should be provided, and APA signals at the CTCF and Cohesin binding sites would also be helpful to represent in parallel.

Response: Thanks for your question and suggestion.

First, the APA signals are quite high mainly because the anchors of significant interactions we identified were TADs, which covered larger genomic regions than loop anchors of CTCF and Cohesin. In detail, we performed APA analysis of long-range (≥ 10 Mb) interactions between condensate-component-enriched TADs using ICE normalized contact matrices at 40kb resolution using the APA function of GENOVA (version 0.9, <https://github.com/robinweide/GENOVA>) with default parameters. We have added details to the figure legend (Revised MS, Page 25) and Methods in the revised manuscript (Revised MS, Page 36).

Second, according to your suggestion, we examined APA signals of interactions between TADs with top 10% binding of CTCF and Cohesin at 40kb resolution in parallel. Interestingly, these interactions also decreased similar with condensate-component-enriched interactions (Response R2 Fig. 14A-B). For this result, the details are discussed as follow:

- a) In our study, what we focused on is the interactions between factor-enriched regions, instead of factor-mediated interactions. Actually, CTCF or Cohesin-enriched regions are overlap with condensate components (Response R2 Fig. 14C). Therefore, it is reasonable that the interactions between their enriched regions also affected by 1,6-HD.
- b) Act as structural factors, CTCF or Cohesin mediated chromatin loops via dimerization^{18,19}. Therefore, interactions mediated by CTCF or Cohesin in form of dimer is expected not be sensitive to 1,6-HD treatment. We examined the effects of 1,6-HD on CTCF or Cohesin(SMC1A)-mediated interactions, which were respectively identified by ChIA-PET and HiChIP experiments in previous studies^{20,21}. Indeed, different with condensate-

component-enriched interactions, these dimerized structural-factor-mediated interactions remained mainly unaffected (Response R2 Fig. 14E-D).

- c) Finally, it is possible that CTCF and Cohesin may also participate in organizing the structure at higher hierarchy (40kb resolution), in combination with other biomolecular condensates. Actually, condensation of CTCF inside nucleus has been observed⁴ and is sensitive to 1,6-HD treatment (Revised MS, Fig. S1B).

Altogether, interactions between CTCF or Cohesin enriched regions were weakened and interactions mediated by CTCF or Cohesin remained mainly unaffected upon 1,6-HD treatment. Therefore, we included the result of CTCF or Cohesin-mediated interactions as a control in new Fig. 7D-E (descriptions in Revised MS, Page 15).

Response R2 Fig. 14: APA analysis of long-range interactions between CTCF or Cohesin-enriched region and CTCF or Cohesin-mediated interactions. (A-B) Left: Aggregate Peak Analysis (APA) analysis of long-range structural factor-enriched interactions at 40kb resolution. Right: Average interactions surrounding the center (<120kb) of structural factor-mediated interactions. All p values were determined by Wilcoxon rank-sum test. **(C)** Upset plot showing the overlap of factor-enriched regions. **(D-E)** Left: APA analysis of structural factor-mediated interactions at 10kb resolution. Right: Average interactions surrounding the center (<30kb) of structural factor-mediated interactions. Pairwise Wilcoxon rank-sum test was performed but no significant changes were detected.

P7 line 1. The 5% and 10% 1,6-HD data were not shown in Figure 1A.

Response: Sorry for the omission. We have removed the citation of Fig .1A after this sentence **(Revised MS, Page 5)**.

P7 line 26. A reference is needed for the statement, "A previous study showed that the reappearance of protein aggregates is likely due to cell shrinkage and increased macromolecular crowding."

Response: Sorry for the omission. We have added this reference **(Revised MS, Page 5)**.

Discussion section, P15, line 41. The authors mentioned that the two preprints also used the 1,6-HD to investigate nuclear condensates' roles in 3D chromatin organization. The author needs to discuss why high concentration and low concentration treatment are opposite? What are the biological implications for this? What are the dangers of using a high concentration of 1,6-HD? Is it also consistent with results presented in two preprints?

Response: Thanks for this advice. The detailed discussion of each point lies as follows:

a) The opposite effect of different conditions of 1,6-HD treatment might occur because high concentration (5% or 10%) of 1,6-HD treatment caused chromatin "freeze"¹⁰, chromatin hyper-condensation, nucleus shrinkage and abnormal protein aggregates in live cells, while low condition and short time treatment does not have those side effects **(Revised MS, Fig. S1)**.

- b) This opposite effect biologically implied that upon high concentration 1,6-HD treatment, the cells may lose membrane function and a potential loss of water within the cells that leads to an increase in macromolecular crowding may have an impact on chromatin behavior.
- c) The danger of using high concentration of 1,6-HD is that high concentration of 1,6-HD severely reduced nucleosome motion, which suggests that high concentration 1,6-HD has a non-specific impact on chromatin mobility. Therefore, high concentration 1,6-HD affected chromatin in a non-specific manner that may have nothing to do with dissolving phase condensates. This non-specific effect makes it difficult to link phase condensate formation to higher order chromatin structure by using this condition of 1,6-HD.
- d) Consistent results were presented in the two preprints. One of the studies observed an increase in long-range interactions upon high concentration of 1,6-HD treatment similar as our observations in long-term exposure to 1,6-HD which caused cell shrinkage. Therefore, their observation in 3D chromatin organization may be a result of cell shrinkage caused by high concentration of 1,6-HD treatment. The other study avoided cell shrinkage by transiently permeabilized cells using Tween 20 and observed moderate change in 3D chromatin organization. The possible reason is that the chromatin may be “frozen” by high concentration of 1,6-HD treatment.

We have improved the Discussion section as follows:

“Two recent studies treating cells with high concentration (5% and 10%, respectively) and long time (15min and 20min, respectively) of 1,6-HD have reported different results in 3D reorganization^{22,23}. One observed moderate 3D reorganization upon treatment²³ whereas the other observed a global 3D reorganization that short-range interactions decreased and long-range interactions increased²², which is consistent with our observation in cells upon 1.5%, long-term (10min and 30min) 1,6-HD treatment and opposite to our results in proper condition. suffered from cell shrinkage and aberrant protein aggregations. The opposite effect of different conditions of 1,6-HD treatment might occur because high concentration(5% or 10%) of 1,6-HD treatment caused chromatin “freeze¹⁰”, chromatin hyper-condensation, nucleus shrinkage and abnormal protein aggregates in live cells, while low condition and short time treatment does not have those side effects (Additional file 1: Fig. S1). These phenotypes are likely caused by a loss of membrane function and a potential loss of water from the cells that leads to an

increase in macromolecular crowding, since the mechanism of 1,6-HD is to dissolve condensates by disrupting hydrophobicity.” (Revised MS, Page 17)

Method section, P24, line 20, Do "RYBP" and "CTCF" were shown in the current work?

Response: We apologize for this mistake. We have corrected this section in the revised manuscript (Revised MS, Page 27).

The authors need to provide enough details about how they perform the puncta analyses.

Response: Thanks for the suggestion. We have added more details in the Materials and Methods part as follows:

“The fixed cells after immunofluorescence were imaged by N-SIM with same parameter across different groups. The puncta number of cells were calculated by the spot module of imaris software (Bitplane). The Slice module was used to measure diameter of puncta. The measured diameter of puncta was input to Surpass module as parameters to calculate the puncta number in selected cell.” (Revised MS, Page 26)

Reviewer #3:

In this paper, the authors use 1,6-HD to explore the possible role of phase condensate formation on chromatin structure. First, they treat ES cells with different amounts of 1,6-HD for different times, and choose a concentration (1.5%) that is less destructive to the cell. They then use HiC to show that 1,6-HD treatment causes changes in TAD structures, including decreasing long range interactions and increasing short range interactions.

Thanks for your comments. We provided detailed point-by-point responses to your comments as follows.

Major concerns

1. The authors talk about phase condensates as though it is completely accepted in the field that these exist in the cell and that they mediate important chromatin interactions. Multiple papers have questioned the existence and/or importance of phase condensates, especially for transcription, chromatin structure or super-enhancer function (see McSwiggen et al. G&D (2019), PMID: 31594803 for an excellent review on this topic). The problem with this, is that even if 1,6-HD dissolves phase condensates in vitro, it is hard to prove what it is actually doing to a living cell, especially if BRD4/Mediator phase condensates turn out to be an artifact.

Response: Thanks for the comments. We agreed that there are many conceptual controversies in this field. One aim of our study was to optimize the tool for disrupting condensates, which may help further understanding the mechanism and roles of biomolecular condensates.

To prove that 1,6-HD dissolves phase condensates in vivo, we performed live-cell fluorescence microscopy of endogenous tagged paraspeckles (indicated by NONO⁹). As a result, we observed the dissolution of paraspeckles in living cells upon short-term 1,6-HD treatment (Response R3 Fig. 1). Besides, we observed a similar time-dependent effect of 1,6-HD with the results in fixed cells (Revised MS, Fig. 1B), where short-term exposure to 1.5% 1,6-HD dissolved condensates and long-term exposure caused re-aggregation. Notably, BRD4 and MED1 were similarly affected by 1,6-HD with those widely accepted condensates (including nucleoli and speckles). These results indicated that 1,6-HD did dissolve phase condensates in vivo.

Response R3 Fig. 1: 1.5% 1,6-HD treatment has a time-dependent effect on paraspeckles in live cell. Live-cell fluorescence microscopy of endogenous tagged paraspeckles (indicated by NONO). The white arrows indicate observed condensates.

So really, this paper is about what 1,6-HD does to chromatin, but if 1,6-HD is a non-specific tool, then it is difficult to draw any biologically meaningful conclusions from the data. What is needed is an independent technique for dissolving phase condensates that is not dependent on 1,6-HD treatment, in order to validate these findings.

Response: Thanks for the in-depth comments. We agreed that 1,6-HD is a non-specific tool as you reminded. Currently, there is no appropriate control that can completely rule out the side effects of 1,6-HD. We still followed reviewers' suggestions and exhausted all available methods to rule out the alternative possibilities. We included proteins that do not form nuclear condensates (Revised MS, Fig. S1B) and 2,5-HD (Revised MS, Fig. S1I-J) as negative controls. Furthermore, we performed live cell imaging to exclude the non-specific impact of 1,6-HD on chromatin mobility and nucleosome clustering (Revised MS, Fig. S1F-H). Importantly, integrated analysis of ChIP-seq shown that the changed 3D chromatin organizations were related with the binding of components in dissolved condensates (Revised MS, Fig. 7). Therefore, it can provide biologically meaningful clues for exploring the relationship between condensates and 3D chromatin organization from our data.

Furthermore, other techniques are used in several studies to illustrate the relationship between phase condensates and 3D chromatin organization that is not dependent on 1,6-HD treatment, including knock down the key components of condensates (Spekle²) or overexpressed mutants that cannot form phase condensates³. These studies have reported similar findings of our work that dissolving condensates resulted in 3D chromatin reorganization. In details, knocking down the core component of speckles reduces chromatin interactions between top 20% active compartments² and the phase-separated ability of OCT4

accounts for TAD reorganizations³.

Altogether, it is biologically meaningful to profile the dynamic of 3D chromatin organization during 1,6-HD treatment, which will provide insights for researching the relationship between condensates and 3D chromatin organization. We also agreed the reviewer's opinion that more appropriate experimental approaches to fully understand the functional role of phase condensates in 3D chromatin organization are further needed.

2. Further to this, the authors have failed to reference an important study in this area (Itoh et al, Life Science Alliance (2021), PMID: 33536240). Itoh et al used Halo-tagged H2B and live imaging to measure chromatin changes and found that chromatin essentially becomes "fixed" at high concentrations of 1,6-HD, similar to the results of long exposure reported here. Itoh et al also find that even at lower doses (2.5% for 5 minutes), nucleosome motion is severely reduced, which suggests that 1,6-HD has a non-specific impact on chromatin mobility. I understand that the authors in this study use an even lower dose of 1,6-HD (1.5%), but it seems quite possible that even at 1.5%, 1,6-HD impacts chromatin mobility in a non-specific manner that has nothing to do with dissolving phase condensates. This non-specific effect could explain the results observed here, but it doesn't accomplish what the authors hope to do, which is to link phase condensate formation to higher order chromatin structure.

Response: Thanks for raising the important concern. We have carefully read the article you mentioned (PMID: 33536240, which was not online when we submitted the manuscript). This study has reported that 5% or higher concentration 1,6-HD treatment for 5min had a side effect of suppressing chromatin motion and hyper-condensing chromatin in live cells¹⁰. We therefore examined the effects of the proper condition we optimized in our work (1.5%, 2min) on chromatin behavior. We performed super-resolution live-cell imaging on stably expressed histone H2B-GFP in HeLa cells and recorded images at 50ms/frame (~300 frame, 15s total) (Response R3 Fig. 2A). Notably, trajectory analysis revealed that although 10%, 5min 1,6-HD drastically suppressed chromatin motion as reported, 1.5%, 2min 1,6-HD did not affect chromatin motion (Response R3 Fig. 2B). Besides, we analyzed the distribution of H2B used the L-function, which quantitates the size of H2B clusters and the degree of accumulation. As a result, 10%, 5min 1,6-HD caused chromatin hyper-condensation as reported, while 1.5%,

2min 1,6-HD exhibited a similar H2B clustering to untreated cells (**Response R3 Fig. 2C**). These results demonstrated that 1.5%, 2min 1,6-HD does not have the side effect of suppressing chromatin motion and hyper-condensing chromatin, which further supported that it is a proper using condition that can be applied for exploring the role of phase condensates in 3D chromatin organization. We have added these results in new **Fig. S1F-H** (descriptions in **Revised MS, Page 7**) and cited this article in our manuscript.

Response R3 Fig. 2: 1.5%, 2min 1,6-HD treatment does not have the side effect of suppressing chromatin motion and hyper-condensing chromatin. (A) Super-resolution live-cell imaging of stably-expressed H2B-GFP. **(B)** Mean square displacement plots indicating chromatin motion (\pm SD among cells). All p values were determined using the Kolmogorov-Smirnov test. **(C)** L-function plot indicating chromatin condensation.

3. Further, if we accept for a moment that proteins such as BRD4 and Mediator actually do form phase condensates in the cell, two recent studies cast doubt on the idea that these proteins have much of an impact on chromatin structure. El Khattabi et al used a degron for Mediator along with Hi-C to show that Mediator is mostly dispensable for chromatin interactions, although they did see a very slight increase in interactions (El Khattabi et al, Cell (2019) PMID: 31402173). More recently, using a degron for BRD4 (as well as BET inhibitors) and a high-resolution 3C method called Capture C, Crump et al showed that enhancer promoter interactions remain intact in the absence of both BRD4 and Mediator, although very slight increases in interactions were also observed (Crump et al, Nat Comm (2021) PMID: 33431820). However, these increased interactions varied between treatments and did not correlate well with gene regulation, so they are likely biologically meaningless. If BRD4 and Mediator form phase condensates in the cell, but degrading these proteins doesn't have an impact on chromatin structure, this casts doubt on whether 1,6-HD mediated chromatin changes are

actually due to disrupting phase condensate formation. Instead, it suggests that the changes observed here that are induced by 1,6-HD could just be an off-target effect.

Response: We appreciate your constructive and in-depth comments. All the works you mentioned are very important findings in the field focusing on the relationship between BRD4/Mediator and Enhancer-Promoter interactions (E-P interactions). Actually, we did observe similar effects on E-P interactions upon 1,6-HD treatment with the degradation of BRD4 or Mediator that E-P interactions remained unaffected (Response R3 Fig. 3A-B). In this study, we focused on higher-order 3D chromatin structures than E-P interactions.

We performed a comparison between the interactions we focused on and E-P interactions. As a result, compared with the E-P interactions, the long-range interactions between condensate-component-enriched genomic regions occurred across larger distances (Response R3 Fig. 3C), and with lower Cohesin binding at both anchors (Response R3 Fig. 3D). Therefore, our findings of changes in the higher-order structures (Compartments and TADs) and long-range interactions do not conflict with the moderate change in E-P interactions reported by these two studies. We have added these results in new Fig. S8A-D (descriptions in Revised MS, Page 19).

Response R3 Fig. 3: Comparison between BRD4- or MED1- enriched interactions and Enhancer-Promoter interactions. (A-B) Left: APA analysis of structural factor-mediated interactions at 10kb resolution. Right: Average interactions surrounding the center (<30kb) of structural factor-mediated interactions. Pairwise Wilcoxon rank-sum test was performed but no significant changes were detected. **(C)** Distance of BRD4- or MED1- enriched interactions and Enhancer-Promoter interactions. **(D)** Cohesin enrichment on anchors of indicated interactions.

The y axis shows reads per kilobase per million reads (RPKM).

Furthermore, El Khattabi et al²⁴ speculated in the Discussion section that the underlying proteins which drive E-P interactions may display a great deal of redundancy, because depletion of RNAPII or Mediator does not overly disrupt chromatin topology²⁴. The redundancy here means that the protein “hubs” containing different types of components may still exist when one particular component was depleted. To preliminarily test this point, we degraded BRD4 using the same condition in the article¹² you mentioned and imaged RNAPII, which is believed to colocalized with BRD4 in phase condensates²⁵. Notably, the number of RNAPII puncta remained unaffected with the absence of BRD4 (**Response R3 Fig. 4**). These results implied that the redundancy of different condensate components may exist. We have added these results in new **Fig. S8G** (descriptions in **Revised MS, Page 19**).

Response R3 Fig. 4: Clues of redundancy between condensate components. Left: SIM images of IF for the proteins indicated by green words. IF for indicated protein is colored green, and signal from DAPI is colored dark blue. Right: Counts of IF puncta within nucleus under different treatments. All p values were determined using the Student’s t test.

Additionally, we used several controls to rule out the alternative possibilities of the off-target effects as described in the answer of the first question. Therefore, dynamic profiles of 3D chromatin organization during 1,6-HD treatment can be used to provide insights for researching the relationship between condensates and 3D chromatin organization.

References:

- 1 Stadhouders, R., Filion, G. J. & Graf, T. Transcription factors and 3D genome conformation in cell-fate decisions. *Nature* **569**, 345-354, doi:10.1038/s41586-019-1182-7 (2019).
- 2 Hu, S., Lv, P., Yan, Z. & Wen, B. Disruption of nuclear speckles reduces chromatin interactions in active compartments. *Epigenetics & chromatin* **12**, 43, doi:10.1186/s13072-019-0289-2 (2019).
- 3 Wang, J. *et al.* Phase separation of OCT4 controls TAD reorganization to promote cell fate transitions. *Cell stem cell*, doi:10.1016/j.stem.2021.04.023 (2021).
- 4 Sabari, B. R., Dall'Agnese, A. & Young, R. A. Biomolecular Condensates in the Nucleus. *Trends in biochemical sciences* **45**, 961-977, doi:10.1016/j.tibs.2020.06.007 (2020).
- 5 Hansen, A. S. *et al.* Distinct Classes of Chromatin Loops Revealed by Deletion of an RNA-Binding Region in CTCF. *Molecular cell* **76**, 395-411.e313, doi:10.1016/j.molcel.2019.07.039 (2019).
- 6 Hansen, A. S., Amitai, A., Cattoglio, C., Tjian, R. & Darzacq, X. Guided nuclear exploration increases CTCF target search efficiency. *Nature chemical biology* **16**, 257-266, doi:10.1038/s41589-019-0422-3 (2020).
- 7 Nair, S. J. *et al.* Phase separation of ligand-activated enhancers licenses cooperative chromosomal enhancer assembly. *Nature structural & molecular biology* **26**, 193-203, doi:10.1038/s41594-019-0190-5 (2019).
- 8 Lin, Y. *et al.* Toxic PR Poly-Dipeptides Encoded by the C9orf72 Repeat Expansion Target LC Domain Polymers. *Cell* **167**, 789-802.e712, doi:10.1016/j.cell.2016.10.003 (2016).
- 9 Yang, L. Z. *et al.* Dynamic Imaging of RNA in Living Cells by CRISPR-Cas13 Systems. *Molecular cell* **76**, 981-997.e987, doi:10.1016/j.molcel.2019.10.024 (2019).
- 10 Itoh, Y. *et al.* 1,6-hexanediol rapidly immobilizes and condenses chromatin in living human cells. *Life science alliance* **4**, doi:10.26508/lsa.202001005 (2021).
- 11 Sabari, B. R. *et al.* Coactivator condensation at super-enhancers links phase separation and gene control. *Science (New York, N.Y.)* **361**, doi:10.1126/science.aar3958 (2018).
- 12 Crump, N. T. *et al.* BET inhibition disrupts transcription but retains enhancer-promoter contact. *Nature communications* **12**, 223, doi:10.1038/s41467-020-20400-z (2021).
- 13 Nora, E. P. *et al.* Targeted Degradation of CTCF Decouples Local Insulation of Chromosome Domains from Genomic Compartmentalization. *Cell* **169**, 930-944.e922, doi:10.1016/j.cell.2017.05.004 (2017).
- 14 Haarhuis, J. H. I. *et al.* The Cohesin Release Factor WAPL Restricts Chromatin Loop Extension. *Cell* **169**, 693-707.e614, doi:10.1016/j.cell.2017.04.013 (2017).
- 15 Quinodoz, S. A. *et al.* Higher-Order Inter-chromosomal Hubs Shape 3D Genome Organization in the Nucleus. *Cell* **174**, 744-757.e724, doi:10.1016/j.cell.2018.05.024 (2018).
- 16 Dali, R. & Blanchette, M. A critical assessment of topologically associating domain prediction tools. *Nucleic Acids Res* **45**, 2994-3005, doi:10.1093/nar/gkx145 (2017).
- 17 Bonev, B. *et al.* Multiscale 3D Genome Rewiring during Mouse Neural Development. *Cell* **171**, 557-572.e524, doi:10.1016/j.cell.2017.09.043 (2017).
- 18 Tang, Z. *et al.* CTCF-Mediated Human 3D Genome Architecture Reveals Chromatin Topology for Transcription. *Cell* **163**, 1611-1627, doi:10.1016/j.cell.2015.11.024 (2015).
- 19 Phillips, J. E. & Corces, V. G. CTCF: master weaver of the genome. *Cell* **137**, 1194-1211, doi:10.1016/j.cell.2009.06.001 (2009).
- 20 Mumbach, M. R. *et al.* HiChIP: efficient and sensitive analysis of protein-directed genome architecture. *Nature methods* **13**, 919-922, doi:10.1038/nmeth.3999 (2016).

- 21 Weintraub, A. S. *et al.* YY1 Is a Structural Regulator of Enhancer-Promoter Loops. *Cell* **171**, 1573-1588.e1528, doi:10.1016/j.cell.2017.11.008 (2017).
- 22 Shi, M. *et al.* Quantifying liquid-liquid phase separation property of chromatin under physiological conditions using Hi-MS and Hi-C. *bioRxiv*, 2020.2012.2007.415489, doi:10.1101/2020.12.07.415489 (2020).
- 23 Ulianov, S. V. *et al.* Suppression of liquid-liquid phase separation by 1,6-hexanediol partially compromises the 3D genome organization in living cells. *Nucleic Acids Res*, doi:10.1093/nar/gkab249 (2021).
- 24 El Khattabi, L. *et al.* A Pliable Mediator Acts as a Functional Rather Than an Architectural Bridge between Promoters and Enhancers. *Cell* **178**, 1145-1158.e1120, doi:10.1016/j.cell.2019.07.011 (2019).
- 25 Cho, W. K. *et al.* Mediator and RNA polymerase II clusters associate in transcription-dependent condensates. *Science (New York, N.Y.)* **361**, 412-415, doi:10.1126/science.aar4199 (2018).

Second round of review

Reviewer 1

In this revised manuscript, the authors have conducted new experiments, including new cell lines and more controls. However, I am not entirely convinced that their work has shown critical biological implications. My major arguments include: 1) The authors aimed to standardize the parameters of 1,6-HD treatment for all cell types. However, I believe that 1.5%, 2min is not a proper condition for all cell types. 2) 1,6-HD is a tool to disrupt condensates in both cytoplasm and nucleus globally, so the link between nuclear condensate formation and higher order chromatin structure can not be addressed in this way.

Reviewer 2

The authors did an excellent job of addressing most of my questions.

I have only one concern about the quality for the imaging data shown in R1 Fig1A; R2 Fig1; R2 Fig2; R2 Fig7; R2 Fig11 R3 Fig4. These imaging data show a large rearrangement of the nucleus as seen in Dapi, the localization of the target proteins (i.e., many of these nuclear proteins appeared in the cytoplasm, and the puncta look like aggregates. They also did not indicate that how many cells were used for quantification.

This is worrisome. These problems need to be addressed before I recommend it for publication.

Reviewer 3

The authors have presented a very careful, and thorough response to my original review. I still have concerns about the possible widespread and non specific effects of 1,6-Hexanediol as a specific research tool, but the additional data and controls provided are quite intriguing. In particular, it is very interesting that RNAPII speckles are retained with BRD4 degradation, and the result of live imaging H2B with 1.5% treatment is very convincing. The new data is so nice, it is a little unfortunate it is in supplemental. This paper does not resolve all of the controversies surrounding phase condensates in transcription, but this is a very carefully done study and I think the work will be very helpful to the field. I think the authors have done an excellent job here.

We appreciate the constructive suggestions and in-depth questions of the three reviewers. We also feel encouraged by the positive comments. We have followed their suggestions and further revised our manuscript (Revised MS, Fig.1A-B, Fig. S1B, Fig. S1J, Fig. S2 and Fig. S8E). In this letter, we provided detailed point-by-point responses to the reviewers' comments together with revised parts of our manuscript.

Reviewer #1:

In this revised manuscript, the authors have conducted new experiments, including new cell lines and more controls. However, I am not entirely convinced that their work has shown critical biological implications. My major arguments include:

Response: Thank you for the comments. We have carefully responded your comments and concerns as follows.

1) The authors aimed to standardize the parameters of 1,6-HD treatment for all cell types. However, I believe that 1.5%, 2min is not a proper condition for all cell types.

Response: Thank you for raising this important concern. We agreed that due to the differences between cell lines, 1.5%, 2 min may not be proper for all cell types and need more test before application.

Nevertheless, we would like to restate that our claim in the manuscript is "1,5%, 2 min is a proper condition of 1,6-HD treatment which is suitable for various cell types", supported by data from mESCs, NPCs and MEFs (Revised MS, Page 8).

Furthermore, since the basic biochemical mechanism of protein condensation is common across cell types¹⁻⁵, this standardized 1,6-HD condition serves as a reference and may help analyze dissolution of condensates in other cell types as well. In fact, we did observe encouraging results in MEFs and NPCs (Revised MS, Additional file 1: Fig. S2). Besides, we also observed a dissolution of paraspeckles upon 1.5%, 2 min 1,6-HD by live cell imaging in HeLa cells (Revised MS, Additional file 1: Fig. S1C). Therefore, when applying 1,6-HD treatment to other cell types, this condition would be a valuable reference.

We added more caveats in the Discussion part in the revised manuscripts as follows:

“We found that 1.5%, 2 min is a proper condition of 1,6-HD usage to dissolve condensates with minimal side effects in mESCs. Since the basic biochemical mechanism of protein condensation is common across cell types^{28,43-46}, this standardized 1,6-HD condition may help analyze dissolution of condensates in other cell types as well, and in fact did show encouraging results in mouse embryonic fibroblasts (MEFs) and neural progenitor cells (NPCs) (Additional file 1: Fig. S2). Besides, we also observed a dissolution of paraspeckles upon 1.5%, 2 min 1,6-HD by live cell imaging in HeLa cells (Additional file 1: Fig. S1C). Therefore, when applying 1,6-HD treatment to other cell types, this condition would be a valuable reference.” (Revised MS, Page 15-16)

2) 1,6-HD is a tool to disrupt condensates in both cytoplasm and nucleus globally, so the link between nuclear condensate formation and higher order chromatin structure can not be addressed in this way.

Response: Thank you for your in-depth comment. We agreed that it is hard to rule out whether the disruption of condensates in cytoplasm or on cell membrane after 1,6-HD treatment has an impact on 3D chromatin organization. Therefore, we replaced the word “nuclear condensates” with “biomolecular condensates” in the whole revised manuscript to illuminate our results more accurately. Besides, what we investigated in this manuscript is “the time-dependent effects of 1,6-HD treatment on biomolecular condensates and 3D chromatin organization”, and we did not claim the causal relationship between nuclear condensate formation and chromatin structure.

Nevertheless, through integrated analysis of ChIP-seq data of components of condensates, we did find chromatin organization changes are related to neighboring nuclear condensates (Revised MS, Fig. 4H, Fig. 5H, Fig. 7, Fig. S4I-L, Fig. S5C-D, Fig. S6F-G, Fig. S7I-L). In particular, the interactions between the condensate-component-enriched regions were greatly affected by 1,6-HD treatment (Revised MS, Fig. 7). Therefore, it still makes sense to discuss the relationship between nuclear condensates and higher order chromatin structure based on this data.

We added more caveats in the Discussion part in the revised manuscripts as follows:

“Since 1,6-HD is a tool to globally disrupt condensates, it is hard to rule out whether the disruption of condensates in cytoplasm or on cell membrane after 1,6-HD treatment has an impact on 3D chromatin organization. Nevertheless, it still makes sense to study the change of 3D chromatin organization upon 1,6-HD treatment. We did find chromatin organization changes are related to neighboring nuclear condensates through integrated analysis of ChIP-seq data of components of condensates and the interactions between the condensate-component-enriched regions were greatly affected by 1,6-HD treatment. This global view of 3D chromatin reorganization after dissolving condensates should help to figure out possible models of the detailed molecular mechanisms and provide clues for further investigation of particular nuclear condensate.” (Revised MS, Page 19)

Reviewer #2:

The authors did an excellent job of addressing most of my questions.

Response: Thanks for your positive comments and constructive suggestions to advance our work. We have carefully responded your comments and questions as follows.

I have only one concern about the quality for the imaging data shown in R1 Fig1A; R2 Fig1; R2 Fig2; R2 Fig7; R2 Fig11 R3 Fig4. These imaging data show a large rearrangement of the nucleus as seen in Dapi, the localization of the target proteins (i.e., many of these nuclear proteins appeared in the cytoplasm, and the puncta look like aggregates.

Response: Thank you for raising this concern. We have improved the quality of imaging data by showing more representative images in the revised manuscript (Response Fig. 1, Revised MS, Fig.1A-B, Fig. S1B, Fig. S1J, Fig. S2 and Fig. S8E) and we would like to add a few points:

- a) For the imaging data you mentioned, we performed super-resolution imaging on fixed cells, which means we separately fixed cells with or without treatment on different plates and performed IF. Therefore, the differences in DAPI distribution are because they are different cells, not because of rearrangement of nucleus upon treatment.
- b) For localization of the target proteins, on one hand, it is normal to detect some signal of nuclear proteins in the cytoplasm because they are synthesized in the cytoplasm and then transported into the nucleus. We improved our figures by equally enhancing the contrast

with same parameter across different groups to reduce low IF signals and found that most nuclear proteins did locate in the nucleus (Response Fig. 1). On the other hand, we found that some nuclear proteins did locate in the cytoplasm in some cell types or under specific conditions. To our surprise, we found that RING1B (key component of polycomb bodies) appeared to locate in the cytoplasm in NPCs (Response Fig. 1A). Besides, we observed that 1.5%, 5min 1,6-HD treatment resulted in translocation of Nucleolin into the cytoplasm (Response Fig. 1E). Importantly, we would like to restate that we only counted puncta inside the nucleus by using the Spots1 module in Surpass (imaris software) and selecting DAPI-stained regions as the “Region of Interest”. We added this detail in the Method part: *“Puncta analysis: The fixed cells after immunofluorescence were imaged by N-SIM with same parameter across different groups. The puncta number of cells were calculated by the spot module of imaris software (Bitplane). The Slice module was used to measure diameter of puncta. The measured diameter of puncta was input to Surpass module as parameters to calculate the puncta number in selected cell. Furthermore, the Spots1 module in Surpass was used to select the “Region of Interest” for calculating the puncta number in nucleus (DAPI-stained regions).”* (Revised MS, Page 28)

- c) For “the puncta look like aggregates”, this is probably because the images were acquired by Structured illumination microscopy (SIM), which mechanically took super-resolution images by reconstructing raw images with N-SIM module. After the reconstruction process, the puncta would look a bit different from the puncta observed under confocal laser scanning microscope.

In a word, we improved the quality of imaging data by showing more representative images, equally enhancing the contrast and adding more details in the Method part in the revised manuscript.

Response Fig. 1: Revised imaging data. SIM images of IF for the proteins indicated by green words in NPCs (A) or mESCs (B-E) upon indicated treatment. IF for indicated protein is colored green, and signal from DAPI is colored dark blue.

They also did not indicate that how many cells were used for quantification.

This is worrisome. These problems need to be addressed before I recommend it for publication.

Response: Sorry for the omission. For each type of condensate under different condition, 30 cells were used for quantification of puncta. We have added this information in the corresponding figure legends (Revised MS, Fig .1, Fig. S1, Fig. S8) and the Methods part:

“Puncta analysis For each type of condensate under different condition, 30 cells were used for quantification of puncta.” (Revised MS, Page 28)

Reviewer #3:

The authors have presented a very careful, and thorough response to my original review. I still have concerns about the possible widespread and non specific effects of 1,6-Hexanediol as a specific research tool, but the additional data and controls provided are quite intriguing. In particular, it is very interesting that RNAPII speckles are retained with BRD4 degradation, and the result of live imaging H2B with 1.5% treatment is very convincing. The new data is so nice, it is a little unfortunate it is in supplemental. This paper does not resolve all of the controversies surrounding phase condensates in transcription, but this is a very carefully done study and I think the work will be very helpful to the field. I think the authors have done an excellent job here.

Response: Thanks for your positive comments and all the constructive suggestions to advance our work, which are really encouraging and helpful to us.

References:

- 1 Sabari, B. R., Dall'Agnesse, A. & Young, R. A. Biomolecular Condensates in the Nucleus. *Trends in biochemical sciences* **45**, 961-977, doi:10.1016/j.tibs.2020.06.007 (2020).
- 2 Banani, S. F., Lee, H. O., Hyman, A. A. & Rosen, M. K. Biomolecular condensates: organizers of cellular biochemistry. *Nature reviews. Molecular cell biology* **18**, 285-298, doi:10.1038/nrm.2017.7 (2017).
- 3 Lyon, A. S., Peeples, W. B. & Rosen, M. K. A framework for understanding the functions of biomolecular condensates across scales. *Nature reviews. Molecular cell biology*, doi:10.1038/s41580-020-00303-z (2020).
- 4 Lafontaine, D. L. J., Riback, J. A., Bascetin, R. & Brangwynne, C. P. The nucleolus as a multiphase liquid condensate. *Nature reviews. Molecular cell biology*, doi:10.1038/s41580-020-0272-6 (2020).
- 5 Alberti, S., Gladfelter, A. & Mittag, T. Considerations and Challenges in Studying Liquid-Liquid Phase Separation and Biomolecular Condensates. *Cell* **176**, 419-434, doi:10.1016/j.cell.2018.12.035 (2019).